# Towards Open-World Grasping with Large Vision-Language Models

**Georgios Tziafas**
Department of Artificial Intelligence
University of Groningen
the Netherlands
`g.t.tziafas@rug.nl`

**Hamidreza Kasaei**
Department of Artificial Intelligence
University of Groningen
the Netherlands
`hamidreza.kasaei@rug.nl`

**Abstract:** The ability to grasp objects in-the-wild from open-ended language instructions constitutes a fundamental challenge in robotics. An open-world grasping system should be able to combine high-level contextual with low-level physical-geometric reasoning in order to be applicable in arbitrary scenarios. Recent works exploit the web-scale knowledge inherent in large language models (LLMs) to plan and reason in robotic context, but rely on external vision and action models to ground such knowledge into the environment and parameterize actuation. This setup suffers from two major bottlenecks: a) the LLM's reasoning capacity is constrained by the quality of visual grounding, and b) LLMs do not contain low-level spatial understanding of the world, which is essential for grasping in contact-rich scenarios. In this work we demonstrate that modern vision-language models (VLMs) are capable of tackling such limitations, as they are implicitly grounded and can jointly reason about semantics and geometry. We propose `OWG`, an open-world grasping pipeline that combines VLMs with segmentation and grasp synthesis models to unlock grounded world understanding in three stages: open-ended referring segmentation, grounded grasp planning and grasp ranking via contact reasoning, all of which can be applied zero-shot via suitable visual prompting mechanisms. We conduct extensive evaluation in cluttered indoor scene datasets to showcase `OWG`'s robustness in grounding from open-ended language, as well as open-world robotic grasping experiments in both simulation and hardware that demonstrate superior performance compared to previous supervised and zero-shot LLM-based methods. Project material is available at https://gtziafas.github.io/OWG_project/.

**Keywords:** Foundation Models for Robotics, Open-World Grasping, Open-Ended Visual Grounding, Robot Planning

## 1 Introduction

Following grasping instructions from free-form natural language in open-ended environments is a multi-faceted problem, posing several challenges to robot agents. Consider the example of Fig. 1: The robot has to decipher the semantics of the user instruction (i.e., *"what would a child want to play with?"*), recognize the appearing objects and ground the target (i.e., the white toy), reason about the feasibility of the grasp to generate an appropriate plan (i.e., first remove the blocking juice box), and finally select a suitable grasp based on

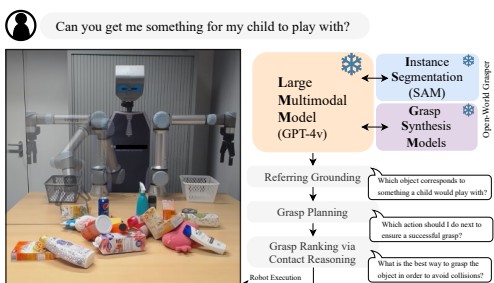

Figure 1: Challenges of open-world grasping tackled with VLMs. The overall pipeline combines VLMs with segmentation and grasp synthesis models to ground open-ended language instructions plan and reason about how to grasp the desired object.

8th Conference on Robot Learning (CoRL 2024), Munich, Germany.

the object geometry and potential collisions. It becomes clear that to deal with the full scope of open-world grasping, agents should integrate high-level semantic with low-level physical-geometric reasoning, while doing so in a generalizable fashion.

In recent years, Large Language Models (LLMs) [1, 2, 3, 4, 5], have emerged as a new paradigm in robotics and embodied AI, due to their emergent general knowledge, commonsense reasoning and semantic understanding of the world [6, 7, 8, 9, 10]. This has led to a multitude of LLM-based approaches for zero-shot robotic task planning [11, 12, 13, 14, 15], navigation [16, 17, 18, 19] and manipulation [20, 21, 22, 23, 24, 25], where the LLM decomposes a high-level language instruction into a sequence of steps, therefore tackling complex, long-horizon tasks by composing primitive skills. However, a notorious limitation of LLMs is their lack of world grounding — they cannot directly reason about the agent and environment physical state [26], and lack deep knowledge when it comes to low-level, physical properties, such as object shapes, precise 3D geometry, contact physics and embodiment constraints [27]. Even when equipped with external visual modules for perceiving the world, the amount of information accessed by the LLM is bottlenecked by the visual model's interface (e.g. open-vocabulary detectors [28, 29, 30] cannot reason about object relations such as contacts). Recently, Large Vision-Language Models (LVLMs) integrate visual understanding and language generation into a unified stream, allowing direct incorporation of perceptual information into the semantic knowledge acquired from language [31, 32, 33, 34]. Preliminary explorations with LVLMs [35] have illustrated two intriguing phenomena, namely: a) by combining LVLMs with segmentation models and constructing suitable visual prompts, LVLMs can unleash extraordinary open-ended visual grounding capabilities [26], and b) effective prompting strategies like chain-of-thought [36] and in-context examples [1] seem to also emerge in LVLMs. Motivated by these results, we perform an in-depth study of the potential contributions of LVLMs in open-ended robotic grasping. In this paper, we propose *Open World Grasper (OWG)*: an integrated approach that is applicable zero-shot for grasping in open-ended environments, object catalogs and language instructions. OWG combines LVLMs with segmentation [37] and grasp synthesis models [38], which supplement the LVLM's semantic knowledge with low-level dense spatial inference. OWG decomposes the task in three stages: open-ended referring segmentation, where the target object is grounded from open-ended language, (ii) grounded grasp planning, where the agent reasons about the feasibility of grasping the target and proposes a next action, and (iii) grasp ranking, where the LVLM ranks grasp proposals generated from the grasp synthesizer based on potential contacts.

In summary, our contributions are threefold: a) we propose a novel algorithm for grasping from open-ended language using LVLMs, b) we conduct extensive comparisons and ablation studies in real cluttered indoor scenes data [39, 40], where we show that our prompting strategies enable LVLMs to ground arbitrary natural language queries, such as open-vocabulary object descriptions, referring expressions and user-affordances, while outperforming previous zero-shot vision-language models by a significant margin, and c) we integrate OWG with a robot framework and conduct experiments both in simulation and in the real world, where we illustrate that LVLMs can advance the performance of zero-shot approaches in the open-world setup.

## 2   Related Works

**Visual Prompting for Vision-Language Models** Several works investigate how to bypass fine-tuning VLMs, instead relying on overlaying visual/semantic information to the input frame, a practise commonly referred to as *visual prompting*. Colorful prompting tuning (CPT) is the first work that paints image regions with different colors and uses masked language models to "fill the blanks" [41]. Other methods try to use CLIP [42] by measuring the similarity between a visual prompt and a set of text concepts. RedCircle [43] draws a red circle on an image, forcing CLIP to focus on a specific region. FGVP [44] further enhances the prompt by specifically segmenting and highlighting target objects. Recent works explore visual prompting strategies for LVLMs such as GPT-4v, by drawing arrows and pointers [35] or highlighting object regions and overlaying numeric IDs [26]. In the same vein, in this work we prompt GPT-4v to reason about visual context while being grounded to specific spatial elements of the image, such as objects, regions and grasps.

**LLMs/LVLMs in Robotics** Recent efforts use LLMs as an initialization for vision-language-action models [45, 46], fine-tuned in robot demonstration data with auxiliary VQA tasks [46, 45, 47]. Such end-to-end approaches require prohibitive resources to reproduce, while still struggling to generalize out-of-distribution, due to the lack of large-scale demonstration datasets. Alternatively, modular approaches invest on the current capabilities of LLMs to decompose language instructions into a sequence of high-level robot skills [48, 11, 12, 22, 14], or low-level Python programs composing external vision and action models as APIs [13, 23, 21, 22, 25, 49]. Such approaches mostly focus on the task planning problem, showcasing that the world knowledge built in LLMs enables zero-shot task decomposition, but require external modules [28, 29, 30, 42] to ground plan steps to the environment and reason about the scene. Recent works study the potential of LVLMs for inherently grounded task planning [27, 50, 51]. In [50], the authors use GPT-4v to map videos of human performing tasks into symbolic plans, but do not consider it for downstream applications. VILA [27] feeds observation images with text prompts to an LVLM to plan without relying on external detectors. However, produced plans are expressed entirely in language and assume an already obtained skill library to execute the plans. MOKA [51] proposes a keypoint-based visual prompting scheme to parameterize low-level motions, but still relies on external vision models to perform grounding, and does not consider referring expressions and clutter.. In our work, we use visual marker prompting to leverage LVLMs for the full stack of the open-world grasping pipeline, including grounding referring expressions, task planning and low-level motion parameterization via grasp ranking.

**Semantics-informed Grasping** Most research on grasping assumes golden grounding, i.e., the target object is already segmented from the input scene. Instead, they focus on proposing 4-DoF grasps from RGB-D views [52, 53, 54, 38, 55, 56, 57], or 6-DoF poses from 3D data [58, 59, 60, 61, 62, 63, 64]. Recently, several works study language-guided grasping in an end-to-end fashion, where a language model encodes the user instruction to provide conditioning for grasping [65, 66, 39]. However, related methods typically train language-conditioned graspers that struggle to generalize outside the training distribution. Another similar line of works is that of task-oriented grasping [67, 68], where recent LLM-based methods [69] exploit the vast knowledge of LLMs to provide additional semantic context for selecting task-oriented grasps, but do not consider the grounding problem, clutter or referring expressions. Further, none of the above approaches consider the planning aspect, typically providing open-loop graspers that do not incorporate environment feedback. In this work, we leverage LVLMs to orchestrate a pipeline for language-guided grasping in clutter, exploiting it's multimodal nature to jointly ground, reason and plan.

# 3 Method

## 3.1 Prerequisites and Problem Statement

**Large Vision-Language Models** VLMs receive a set of RGB images of size $H \times W$: $\mathcal{I}_{1:M}$, $\mathcal{I} \in \mathbb{R}^{H \times W \times 3}$ and a sequence of text tokens $\mathcal{T}$, and generate a text sequence $\mathcal{Y}$ of length $L$: $\mathcal{Y} \doteq w_{1:L} = \{w_1, \ldots, w_L\}$ from a fixed token vocabulary $w_i \in \mathcal{W}$, such that: $\mathcal{Y} = \mathcal{F}(\mathcal{I}_{1:M}, \mathcal{T})$. The images-text pair input $\mathcal{X} = \langle \mathcal{I}_{1:M}, \mathcal{T} \rangle$ is referred to as the *prompt*, with the text component $\mathcal{T}$ typically being a user instruction or question that primes the VLM for a specific task.

**Grasp Representations** We represent a grasp via an end-effector gripper pose $\mathcal{G}$, with $\mathcal{G} \in \mathbb{R}^4$ for 4-DoF and $\mathcal{G} \in \mathbb{R}^6$ for 6-DoF grasping. Such representation contains a 3D position and either a yaw rotation or a full SO(3) orientation for 4-DoF and 6-DoF respectively. 4-DoF grasps assume that the approach vector is calibrated with the camera extrinsics, and hence can be directly drawn as rectangles in the 2D image plane (see bottom of Fig. 2), which happens to be a favorable representation for VLMs, as grasp candidates can be interpreted as part of the input image prompt. A motion primitive is invoked to move the arm to the desired gripper pose $\mathcal{G}$, e.g. via inverse-kinematics solvers. [1]

---

[1]More sophisticated motion planning algorithms, e.g. with integrated obstacle avoidance, can be utilized orthogonal to our approach.

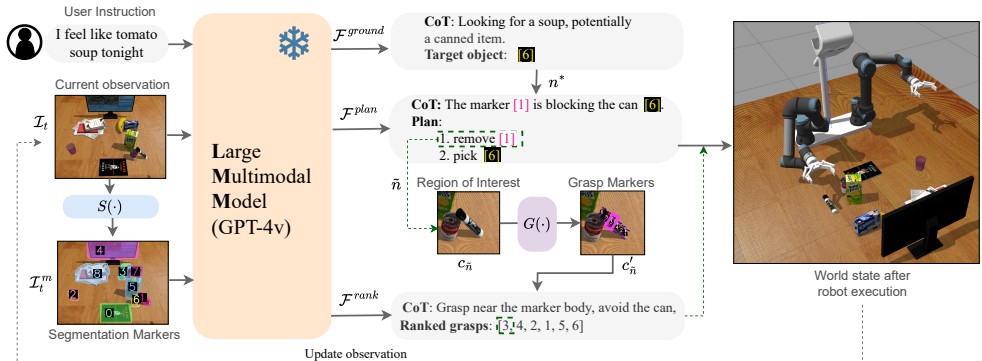

Figure 2: **Overview of OWG:** Given a user instruction and an observation, OWG first invokes a segmentation model to recover pixel-level masks, and overlays them with numeric IDs as visual markers in a new image. Then the VLM subsequently activates three stages: (i) grounding the target object from the language expression in the marked image, (ii) planning on whether it should grasp the target or remove a surrounding object, and (iii) invoking a grasp synthesis model to generate grasps and ranking them according to the object's shape and neighbouring information. The best grasp pose (highlighted here in pink - not part of the prompt) is executed and the observation is updated for a new run, until the target object is grasped. Best viewed in color and zoom.

**Problem Statement** Given an RGB-D observation $\mathcal{I}_t \in \mathbb{R}^{H \times W \times 3}$, $\mathcal{D}_t \in \mathbb{R}^{H \times W}$ and an open-ended language query $\mathcal{T}$, which conveys an instruction to grasp a target object, the goal of OWG is to provide a policy $\pi(a_t \mid \mathcal{I}_t, \mathcal{D}_t, \mathcal{T})$. Assuming $n \in \{1, \ldots, N\}$ the $N$ objects that appear in the scene and $n^*$ the target object, then at each time step $t$, the policy outputs a pose for grasping an object: $a_t = G_t(n)$, $G_t(n) = G(n, \mathcal{I}_t, \mathcal{D}_t)$, $t = 1, \ldots, T$, where the last step $T$ always maps to grasping the target object: $a_T = G_T(n^*)$. We refer to the function $G$ as the *grasp generation* function, which corresponds to a pretrained grasp synthesis network from RGB-D views [38] [2] We note that our policy $\pi$ outputs directly the actual gripper pose $\mathcal{G} = G(n)$, and the object-centric abstraction $n$ is used implicitly (details in next sections).

We wish to highlight that in most grasp synthesis pipelines [38, 55, 53, 57, 56], it's always $T = 1$ and $a_1 = G_1(n^*)$, which corresponds to an open-loop policy attempting to grasp the object of interest once. Our formulation for $T > 1$ allows the VLM to close the loop by re-running after each step, which enables visual feedback for planning and recovery from failures / external disturbances.

## 3.2   Pipeline Overview

OWG combines VLMs with pretrained 2D instance segmentation and grasp synthesis models. Segmentation methods like SAM [37] and its variants [70, 71] have demonstrated impressive zero-shot performane. Similarly, view-based grasp synthesis networks [56, 55, 38, 53, 57] have also shown to be transferable to unseen content, as they are trained without assumptions of objectness or semantics in their training objectives. The zero-shot capabilities of these models for low-level dense spatial tasks is complementary to the high-level semantic reasoning capabilities of VLMs, while both use images as the underlying representation, hence offering a very attractive coupling for tackling the open-world grasping problem. The overall pipeline can be decomposed in three subsequent stages: (i) open-ended referring segmentation, (ii) grounded grasp planning, and (iii) grasp generation and ranking. A schematic of OWG is shown in Fig. 2 and described formally in Algorithm 1. Prompt implementation details can be found in Appendix A.

**Open-ended referring segmentation** In this stage, the target object of interest must be segmented from the input RGB image $\mathcal{I}_t$ given the instruction $\mathcal{T}$. To enable this, we first run our segmentation model $S : \mathbb{R}^{H \times W \times 3} \to \{0, 1\}^{H \times W}$ and then draw the $N$ generated masks $M_{1:N} = S(\mathcal{I}_t)$ with additional visual markers in a new frame $\mathcal{I}_t^m$. This step aims to exploit the VLM's OCR capabilities

---
[2]Other point-cloud [59] or voxel-based [62] methods for 3D grasp generation can be utilized orthogonal to our approach, which uses single RGB-D view.

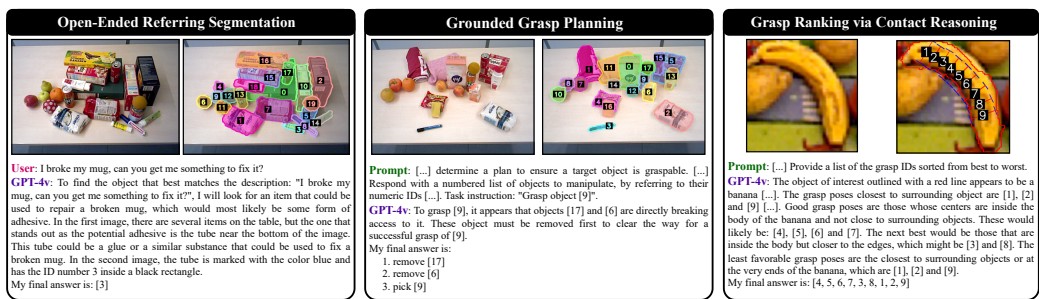

Figure 3: Example GPT-4v responses (from left to right): a) Open-ended referring segmentation, i.e., grounding, b) Grounded grasp planning, and c) Grasp ranking via contact reasoning. We omit parts of the prompt and response for brievity. Full prompts in Appendix A and more example responses in Appendix E.

and link each segment in the frame with a unique ID that the VLM can use to refer to it. After augmenting the image with visual markers, we pass the prompt $< \mathcal{I}_t, \mathcal{I}_t^m, \mathcal{T} >$ to the VLM. We refer to this VLM generation as $\mathcal{F}^{ground}$, such that: $n^* = \mathcal{F}^{ground}(\mathcal{I}_t, \mathcal{I}_t^m, \mathcal{T})$ where $n^*$ the target object and $M_{n^*}$ its segmentation mask. We note that $\mathcal{T}$ can contain free-form natural language referring to a target object, such as open object descriptions, object relations, affordances etc.

**Grounded grasp planning** This stage attempts to leverage VLM's visual reasoning capabilities in order to produce a plan that maximizes the chances that the target object $n*$ is graspable. If the target object is blocked by neighboring objects, the agent should remove them first by picking them an placing them in free tabletop space. Similar to [27], we construct a text prompt that describes these two options (i.e., *remove* neighbor or *pick* target) as primitive actions for the VLM to compose plans from. We provide the marked image $\mathcal{I}_t^m$ together with the target object $n^*$ (from the previous grounding stage) to de-

---

**Algorithm 1: Open-World Grasper (OWG)**

---

**Require:** Initial observation $(\mathcal{I}_1, \mathcal{D}_1)$, language instruction $\mathcal{T}$, segmentor $S(\cdot)$, grasp generator $G(\cdot)$, VLMs $\mathcal{F}^{ground}, \mathcal{F}^{plan}, \mathcal{F}^{rank}$
**Ensure:** $n^* \neq \tilde{n}$
    $t \leftarrow 1$
    **while** $n^* \neq \tilde{n}$ **do**
        Generate segmentation masks $M_{1:N}$ with $S(\mathcal{I}_t)$
        Draw visual markers from $M_{1:N}$ in a new frame $\mathcal{I}_t^m$
        $n^* \leftarrow \mathcal{F}^{ground}(\mathcal{I}_t, \mathcal{I}_t^m, \mathcal{T})$   ▷ Object of interest
        $\tilde{n} \leftarrow \mathcal{F}^{plan}(\mathcal{I}_t^m, n^*)[0]$   ▷ Next object to grasp
        $\mathcal{G}_{1:K} \leftarrow G(\mathcal{I}_t, \mathcal{D}_t, M_{\tilde{n}})$   ▷ Grasp generation
        Crop RoI and draw grasps $c_{\tilde{n}'}$ from $\mathcal{I}_t, M_{\tilde{n}}, G_{1:K}$
        $\mathcal{G}'_{1:K} \leftarrow \mathcal{F}^{rank}(c_{\tilde{n}'})$   ▷ Grasp ranking
        Execute grasp $\mathcal{G}'_1$
        $t \leftarrow t + 1$   ▷ Update observation $\mathcal{I}_t, D_t$
    **end while**

---

termine a plan: $p_{1:T} = \mathcal{F}^{plan}(\mathcal{I}_t^m, n^*)$, $p_\tau \in \{1, \dots, N\}$. Each $p_\tau$ corresponds to the decision to grasp the object with marker ID $n \in \{1, \dots, N\}$. As motivated earlier, in order to close the loop, we take the target of the first step of the plan $\tilde{n} = p_1$ and move to the grasping stage of our pipeline.

**Grasp generation and ranking** After determining the current object to grasp $\tilde{n}$, we invoke our grasp synthesis model $G$ to generate grasp proposals. To that end, we element-wise multiply the mask $M_{\tilde{n}}$ with the RGB-D observation, thus isolating only object $n^*$ in the input frames: $\tilde{\mathcal{I}}_t = \mathcal{I}_t \odot M_{\tilde{n}}$, $\tilde{\mathcal{D}}_t = \mathcal{D}_t \odot M_{\tilde{n}}$. The grasp synthesis network outputs pixel-level quality, angle and width masks which can be directly transformed to 4-DoF grasps $\mathcal{G}_{1:K} = G(\tilde{\mathcal{I}}_t, \tilde{\mathcal{D}}_t)$ [56, 55, 38], where $K$ the total number of grasp proposals. Then, we crop a small region of interest $c_{\tilde{n}}$ around the bounding box of the segment in the frame $\mathcal{I}_t$, from its mask $M_{\tilde{n}}$. We draw the grasp proposals $\mathcal{G}_{1:K}$ as 2D grasp rectangles within the cropped image $c_{\tilde{n}}$ and annotate each one with a numeric ID marker, similar to the grounding prompt. We refer to the marked cropped frame as $c'_{\tilde{n}}$. Then, we prompt the VLM to rank the drawn grasp proposals: $\mathcal{G}'_{1:K} = \mathcal{F}^{rank}(c'_{\tilde{n}})$ where the prompt instructs the VLM to rank based on each grasp's potential contacts with neighboring objects. Finally, the grasp ranked best by the VLM $\mathcal{G}'_1$ is selected and sent to our motion primitive for robot execution.

## 4 Experiments

In this section, we compare the open-ended grounding capabilities of OWG vs. previous zero-shot methods in indoor cluttered scenes (Sec. 4.1). Then, we demonstrate its potential for open-world

grasping both in simulation and in hardware (Sec. 4.2). Finally, we investigate the effect of several components of our methodology via ablation studies (Sec. 4.3).

## 4.1 Open-Ended Grounding in Cluttered Scenes

In order to evaluate the open-ended potential of OWG for grounding, we create a small subset of OCID-VLG test split [39], which we manually annotate for a

| Method | Found. Model | Name | Attribute | Spatial Relation | Visual Relation | Semantic Relation | Affordance | Multi-hop | Avg. |
|---|---|---|---|---|---|---|---|---|---|
| ReCLIP [72] | CLIP [42] | 71.4 | 57.7 | 27.3 | 47.4 | 46.2 | 62.5 | 20.8 | 47.6±17.0 |
| RedCircle [43] | CLIP [42] | 52.4 | 53.9 | 18.2 | 42.1 | 46.2 | 18.9 | 12.5 | 34.8±16.4 |
| FGVP [44] | CLIP [42] | 50.0 | 53.9 | 33.3 | 36.9 | 53.8 | 43.8 | 29.1 | 43.0±9.3 |
| FGVP* [44] | CLIP [42] | 65.7 | 65.4 | 33.3 | 42.1 | 69.2 | 56.2 | 29.1 | 51.8±15.4 |
| QWEN-VL-2 [31] | QWEN [31] | 64.3 | 60.9 | 52.4 | 44.0 | 47.1 | 11.9 | 42.1 | 46.1±15.9 |
| SoM [26] | GPT-4v [73] | 54.8 | 42.3 | 54.6 | 57.9 | 53.9 | 62.5 | 45.8 | 53.1±6.4 |
| OWG (Ours) | GPT-4v [73] | **85.7** | **80.8** | **75.8** | **73.7** | **76.9** | **93.8** | **79.2** | **80.8**±6.4 |

Table 1: Zero-shot referring segmentation - mIoU(%) results per language instruction type for cluttered indoor scenes from OCID [40].

broad range of grasping instructions. As we strive for zero-shot usage in open scenes, we mostly experiment with previous visual prompting techniques for large-scale VLMs, such as CLIP [43, 44, 72], as well as the recent Set-of-Mark prompting methodology for GPT-4v [26], which constitutes the basis of our method. We also include comparisons with open-source visually-grounded LVLM QWEN-VL-2 [31]. Please see Appendix C for details on the test dataset, baseline implementations and more comparative ablations and qualitative results.

We observe that both CLIP-based visual prompting techniques and open-source LVLMs are decent in object-based but fail to relate objects from the visual prompts. Even GPT-4v-based SoM prompting method is not directly capable of handling cluttered tabletop scenes from depth cameras, as is evident by the 53.1% averaged mIoU across all query types. Overall, our OWG-grounder achieves an averaged mIoU score of 80.8%, which corresponds to a 27.7% delta from the second best approach. Importantly, OWG excels at semantic and affordance-based queries, something which is essential in human-robot interaction applications but is missing from modern vision-language models. We identify two basic failure modes: a) the LVLM confused the target description with another object, e.g. due to same appearance or semantics, and b) the LVLM reasons correctly about the object and where it is roughly located, but chooses a wrong numeric ID to refer to it.

## 4.2 Open-World Grasping Robot Experiments

In this section we wish to evaluate the full stack of OWG, incl. grounding, grasp planning and grasp ranking via contact reasoning, in scenarios that emulate open-world grasping challenges. To that end, we conduct experiments in both simulation and in hardware, where in each trial we randomly place 5-15 objects in a tabletop and instruct the robot to grasp an object of interest. We conduct trials in two scenarios, namely: a) **isolated**, where all objects are scattered across the tabletop, b) **cluttered**, where objects are tightly packed together leading to occlusions and rich contacts. We highlight that object-related query trials contain distractor objects that share the same category with the target object.

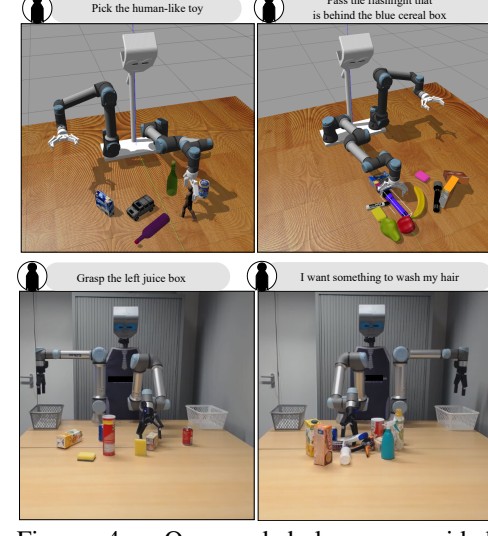

Figure 4: Open-ended language-guided grasping trials in Gazebo *(top)* and real robot *(bottom)*, in isolated *(left column)* and cluttered *(right column)* scenes.

**Baselines** We compare with two baselines, namely: a) **CROG** [39], an end-to-end referring grasp synthesis model trained in OCID [40] scenes, and b) **SayCan-IM** [12], an LLM-based zero-shot planning method that actualizes embodied reasoning via chaining external modules for segmentation, grounding and grasp synthesis, while reasoning with LLM chain-of-thoughts [74]. Our choice of baselines aims at showing the advantages of using an

LVLM-based method vs. both implicit end-to-end approaches, as well as modular approaches that rely solely on LLMs to reason, with visual processing coming through external tools. See details in baseline implementations in Appendix B.

**Implementation** Our robot setup consists of two UR5e arms with Robotiq 2F-140 parallel jaw grippers and an ASUS Xtion depth camera. We conduct 50 trials per scenario in the Gazebo simulator [75], using 30 unique object models. For real robot experiments, we conduct 6 trials per scenario having the initial scenes as similar as possible between baselines. In both SayCan-IM and our method, Mask-RCNN [76] is utilized for 2D instance segmentation while

| Setup | CROG [39] | | SayCan-IM [12] | | OWG (Ours) | |
|---|---|---|---|---|---|---|
| | *seen* | *unseen* | *seen* | *unseen* | *seen* | *unseen* |
| Simulation (×50) | | | | | | |
| *-Isolated* | 66.0 | 36.0 | 62.0 | 60.0 | 78.0 | 82.0 |
| *-Cluttered* | 38.0 | 22.0 | 48.0 | 56.0 | 62.0 | 66.0 |
| Real-World (×6) | | | | | | |
| *-Isolated* | 50.0 | 16.6 | 66.6 | 33.3 | 83.3 | 66.6 |
| *-Cluttered* | 16.6 | 0.0 | 16.6 | 16.6 | 50.0 | 50.0 |

Table 2: Averaged success rates (%) over simulated and real-world grasping trials. The × represents number of trials per cell.

GR-ConvNet [38] pretrained in Jacquard [52] is used as the grasp synthesis module. Our robotic setup is illustrated in Fig. 4, while more details can be found in Appendix B. To investigate generalization performance, all method are evaluated in both scenarios, in two splits: *(i)* **seen**, where target objects and queries are present in the method's training data or in-context prompts, and *(ii)* **unseen**, where the instruction refers to objects that do not appear in CROG's training data or SayCan-IM's in-context prompts. Averaged success rate per scenario is reported, where a trial is considered successful if the robot grasps the object and places it in a pre-defined container position.

**Results** We observe that the supervised method CROG struggles when used at unseen data, in both scenarios. In contrary, both SayCan-IM and OWG demonstrate immunity to seen/unseen objects, illustrating the strong zero-shot capabilities of LLM-based approaches, which can naturally generalize the concepts of object categories/attributes/relations from language. SayCan-IM is limited by the external vision models and hence struggles in cluttered scenes, where its detector sometimes fails to perceive

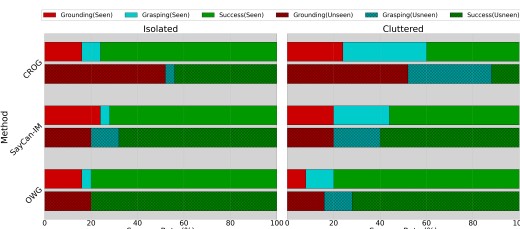

Figure 5: Distribution of failures across grounding and grasping in Gazebo grasping trials for isolated *(left)* and cluttered *(right)*. OWG improves performance across both modes in both setups and test splits.

the target object, resulting in lower final success rates compared to OWG, especially in the real-world experiments. OWG consistently outperforms both baselines both in simulation and in the real robot, with an $\sim 15\%$ and $\sim 35\%$ improved averaged success rate respectively. In Fig. 5, we illustrate the decomposition of failures across grounding and grasping in our baselines for 25 Gazebo trials per scenario, where we automatically test for the target object's grounding results alongside success rate. We observe that OWG consistently reduces the error rates in both grasping and grasping compared to the baselines in all scenarios and test splits. We believe that these results are encouraging for the future of LVLMs in robot grasping.

### 4.3 Ablation Studies

In out ablations we wish to answer the following questions: a) What is the bottleneck introduced by the segmentation model in the open-ended grounding performance?, b) What are the contributions of all the different visual prompt elements considered in our work?, and c) What is the contribution of the LVLM-based grasp planning and ranking in robot grasping experiments? The grounding ablations for the first two questions are organized in Table 3, while for the latter in Table 4.

**Instance segmentation bottleneck** We compare the averaged mIoU of our OWG grounder in a subset of our OCID-VLG evaluation data for three different segmentation methods and ground-truth masks. We employ: a) SAM [37], b) the RPN module of the open-vocabulary detector ViLD [28], and c) the RGB-D two-stage instance segmentation method UOIS [77], where we also provide the depth data as part of the input. ViLD-RPN and UOIS both achieve a bit above 70%, which is a

$\sim 15\%$ delta from ground-truth masks, while SAM offers the best baseline with a $10.8\%$ delta from ground-truth. Implementation details and related visualizations in Appendix C.

**Visual prompt components** Visual prompt design choices have shown to significantly affect the performance of LVLMs. We ablate all components of our grounding prompt and observe the contribution of each one via its averaged mIoU in the same subset as above (see details in Appendix A.2). The most important prompt component is the reference image, provided alongside the marked image. Due to the high clutter of our test scenes, simply highlighting marks and label IDs in a single frame, as in SoM [26] hinders the recognition capabilities of the LVLM, with a mIoU drop from $86.6\%$ to $23.2\%$. Further decluttering the marked image

| Method | mIoU (%) |
|---|---|
| OWG (w/ *Ground-Truth Mask*) | 86.6 |
| -w/o reference | 23.2 |
| -w/o number overlay | 54.6 |
| -w/o high-res | 61.3 |
| -w/o self-consistency | 70.9 |
| -w/ box | 74.6 |
| -w/o CoT prompt | 77.6 |
| -w/o mask fill | 81.1 |
| SAM [37] | 75.8 |
| ViLD-RPN [78] | 72.9 |
| UOIS [77] | 71.1 |

Table 3: Grounding ablation studies.

also helps, with overlaying the numeric IDs, using high-resolution images and highlighting the inside of each region mask being decreasingly important. Surprisingly, also marking bounding boxes leads to a 12% mIoU drop compared to avoiding them, possibly due to occlusions caused by lots of boxes in cluttered areas. Finally, self-consistency and chain-of-thought prompting components that were added also improve LVLM's grounding performance by $\sim 16$ and $10\%$ respectively, by ensembling multiple responses and enforcing step-by-step reasoning.

**Grasp-Related Ablations** We quantify the contribution of our grasp planning and ranking stages in the open-world grasping pipeline, by replicating trials as in the previous section and potentially skipping one or both of these stages. As we see in Table 4, the effect of these components is not so apparent in isolated scenes, as objects are not obstructed by surroundings and hence most proposed

| Method | Isolated | Cluttered |
|---|---|---|
| OWG | 84.0 | 68.0 |
| -w/o planning | 80.0 | 46.0 |
| -w/o grasp ranking | 82.0 | 60.0 |
| -w/o both | 80.0 | 42.0 |

Table 4: Averaged success rates (%) over 50 simulated grasping trials per scenario.

grasps are feasible. The effect becomes more prominent in the cluttered scenario, where the lack of grasp planning leads to a success rate decrease of $22\%$. This is because without grasp planning the agent attempts to grasp the target immediately, which almost always leads to a collision that makes the grasp fail. Grasp ranking is less essential, as a lot of contact-related information is existent in the grasp quality predictions of our grasp synthesis network. However, it still provides an important boost in final success rate ($8\%$ increase). When skipping both stages, the agent's performance drops drastically in cluttered scenes, as it is unable to recover from grasp failures, and hence always fails when the first attempted grasp was not successful.

## 5 Conclusion, Limitations & Future Work

In this paper we introduce OWG, a novel system formulation for tackling open-world grasping. Our focus is on combining LVLMs with segmentation and grasp synthesis models, and visually prompt the LVLM to ground, plan and reason about the scene and the object grasps. Our works sets a foundation for enabling robots to ground open-ended language input and close-the-loop for effective grasp planning and contact reasoning, leading to significant improvements over previous zero-shot approaches, as demonstrated by empirical evaluations, ablation studies and robot experiments.

**Limitations** First, as OWG is a modular approach, it suffers from error cascading effects introduced by the segmentor and grasp synthesis models. However, improvements in these areas mean direct improvement to the OWG pipeline. Second, we currently use 4-DoF grasps to communicate them visually to GPT-4v, which constrains grasping to single view. In the future we would like to integrate 6-DoF grasp detectors and explore new prompting schemes to aggregate and rank grasp information visually. Third, our results suggest that LVLMs still struggle to ground complex object relationships. More sophisticated prompting schemes beyond marker overlaying, or instruct-tuning in grasp-related data, might be a future direction for dealing with this limitation.

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

# Towards Open-World Grasping with Large Vision-Language Models

**Supplementary Material**

## A   LVLM prompts

### A.1   Prompts

Prompts for the three use cases considered in this work, namely: **open-ended referring segmentation**, **grounded grasp planning**, and **grasp ranking** can be found below.

- **Open-ended referring segmentation:** referring_segmentation.txt
  Takes an observation image, a marked copy with highlighted instance masks and label IDs and an open-ended language query referring to a particular object instance, and outputs the label ID of the referred object. The LVLM is encouraged to provide chains-of-thought reasoning in cases where the input query contains complex expressions that involve multiple object and spatial relations.

- **Grounded grasp planning:** grasp_planning.txt
  Takes the marked image and the label ID of the target object to be grasped, and outputs a plan to ensure the target object will be graspable. The plan consists of `remove` actions for blocking objects and a final `pick` action for the target.

- **Grasp ranking:** grasp_ranking.txt
  Takes a cropped bounding box image around the next object to be picked, marked with grasp proposals from a 4-DoF grasp synthesis model and a set of label IDs, and outputs a sorted list of grasp IDs, from most to least confidence for a successful grasp. The LVLMs is encouraged to reason about the object shape and their neighbouring objects before producing a final ranking.

### A.2   Visual Prompt Design

In the following, we summarize the key visual prompting elements that were used for prompting the LVLM in the context of OWG:

**Clarity of visual markers** The most common failure mode of visual marker prompting with GPT-4v is that it sometimes struggles to discriminate which ID corresponds to what segment. Especially in cluttered scenes, label IDs might severely overlap within small frame regions. Several techniques can assist in making the markers more clear to the LVLM: a) we adopt the algorithm of [1] for overlaying numeric IDs within the frame with minimal overlap, b) we paint both the internal of each segment's mask and its ID with the same unique color (colors are chosen to be visually distinguishable), and c) increase the resolution of the marked image and the size layout of the markers.

**Reference Image** If not highlighting the internal of each segment, GPT-4v sometimes refers to regions with wrong IDs, especially in highly cluttered scenes. But if the masks are highlighted with high opacity, then the appearance of the object becomes less visible and GPT-4v struggles to recognize it. We propose a technique to ameliorate this is by passing both the original (reference) and the marked image and constructing a text prompt that explains that the latter corresponds to annotated segments of the first.

8th Conference on Robot Learning (CoRL 2024), Munich, Germany.

**Chain-of-thoughts** Chain-of-Thought (CoT) prompting is a well-established methodology for guiding LLMs to perform multi-step reasoning and reduce hallucinations [2]. We find that LVLMs share similar properties and prompting them to reason about their final answer before producing it can robustify the response quality. For grounding, we ask GPT-4v to decompose the input instruction in steps and refer to all intermediate referenced objects. For grasp planning, we ask it to explicitly mention all object IDs that are blocking the target object, before producing a plan. For grasp ranking, we decompose the prompt in three steps: (i) identify the category of the target object and provide a general description of what constitutes a good grasp for it given its shape, (ii) list the grasp IDs that will most likely lead to contact with neighboring objects, and (iii) rank the grasp IDs based on the previous two steps.

**Self-consistency** Even with zero temperature, we observe that the outputs of GPT-4v are not always reproducible. We find that sometimes GPT-4v might produce different responses at different runs, even with exactly the same prompt. In an attempt to reduce the effect of this phenomenon and robustify LVLM outputs, we use the self-consistency method developed for LLMs [3]. In particular, we ask GPT-4v to provide multiple responses, parse each one separately and then perform majority voting to determine the most consistent output.

## B  Robot experiments

### B.1  Setups

Our object catalog for seen/unseen trials is shown in Fig. 1. In Gazebo, isolated scenarios are generated by ensuring all spawned objects have a fixed 3D distance, while in cluttered scenarios we ensure contact between the target object and neighbouring objects, by first spawning the target and then sampling different poses for other object models around it. In real-robot experiments, we manually setup the scenes while making sure to replicate the setup as close as possible for fair comparisons between baselines. In all trial scenes that contain distractor objects, the user instruction refers to some property that disambiguates the target instance from other objects of the same category, using names, attributes and spatial relations. We also conduct experiments without distractors for affordance-based queries, which require semantic reasoning to be correctly grounded.

For real robot experiments, we use the default `torchvision` implementation of Mask-RCNN, with the model weights provided by PyTorch Hub, fine-tuned in a few annotated scenes captures from our robot setup. For grasp synthesis, we generate a top-down orthographic projection of the scene, both for color and for depth (i.e. reverse depth - heightmap). This is the input we pass to the pretrained GR-ConvNet. In order to align regions from the 2D frame where Mask-RCNN provides segmentations and the orthograpic projection where our grasp synthesis model provides grasp poses, we use the Hungarian matching algorithm to match the centers of outputs from both models, after projected to 3D and

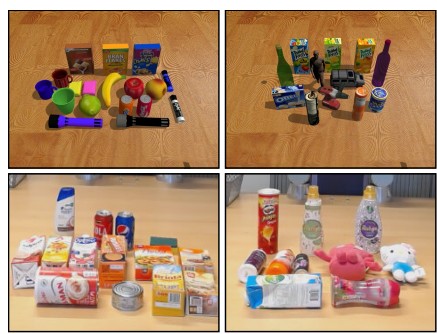

Figure 1: Seen *(left column)* and unseen *(right column)* object used in our robot experiments in Gazebo *(top)* and the real world *(bottom)*.

transformed to a world reference frame (robot base), using 3D euclidean distance as the cost function.

### B.2  Baseline Implementation

**CROG** CROG receives an single $448 \times 448$ RGB view and a natural language query, and provides both an instance segmentation mask for the target object, as well as a set of 4-DoF grasp proposals, assuming that the gripper approaches the object aligned with the perspective of the camera. We use the checkpoint provided by the original paper, trained in the multiple split of OCID-VLG dataset, which contains 90k scene-query-grasp data from around 1,000 unique scenes from 31

| Name | Attribute | Spatial Rel. | Visual Rel. | Sem. Rel. | Multi-hop | Affordance | Total |
|------|-----------|--------------|-------------|-----------|-----------|------------|-------|
| 42 | 26 | 33 | 19 | 13 | 24 | 16 | 173 |

Table 1: Number of samples in grounding evaluation dataset.

object categories. The model uses CLIP's pretrained ResNet-50 visual and BERT text encoders, but fine-tunes them end-to-end in OCID scenes for joint grounding and grasp synthesis tasks.

**SayCan-IM** Our SayCan-IM baseline follows the implementation publicly released by the SayCan work [4], which can be found in this HTTP URL. In particular, the pipeline uses the ViLD [5] open-vocab object detector to turn the input observation image into a list of object names and then lets the LLM generate a sequence of pick-and-place actions to perform in order to solve the task given by the user. We made the following modifications to the above baseline:

1. In the original implementation, the robot only has access to a `pick_and_place` skill, and the output plan is confined to only selecting what objects to pick and where to place them (based on the detected object list from ViLD). In our implementation, we also provide a `visual_grounding` tool, which lets the LLM invoke CLIP [6] to rank a list of candidate objects with a given text description and select the most similar one. This is to allow the LLM reason about attribute concepts besides object category (e.g. *"get the blue mug"*).

2. Besides the names of the appearing objects, we also provide their bounding box coordinates, as detected by ViLD, in `x1y1x2y2` format in the prompt. This was introduced in order to enable the LLM to also reason about the locations of objects and resolve spatial relation queries, as well as reason about the feasibility of grasping objects by checking whether their bounding boxes overlap.

3. We replace the `pick_and_place` primitive skill with two distinct skills: `remove` and `pick`. The first skill corresponds to removing a blocking object in order to clear the path for grasping the target. The second skill corresponds to picking the target object that the user requested. Both skills use GR-ConvNet [7] under-the-hood to sample grasp proposals, select the one with higher predicted grasp quality, and use an IK solver to control the robot arm.

4. We used the observe-reason-act prompting style first introduced by Inner Monologue [8] and later improved by ReAct [9]. Unlike the vanilla implementation, which simply produces a plan of steps without feedback, with this technique we let the LLM plan one step at a time, and integrate feedback from the environment (e.g. CLIP outputs, grasp failures etc.) before planning again.

The system prompt and in-context examples used in our SayCan-IM baseline are shown in Fig. 2. As we mention in our main paper, for the real robot experiments, we replace ViLD-RPN with a Mask-RCNN [10] for instance segmentation, and use CLIP with prompts for all object used in experiments to recognize the categories and provide the object list state to the LLM.

## C  Offline grounding experiments

### C.1  OCID Dataset Details

We manually annotate 173 images from OCID dataset with the following query types: a) **name** (open-vocabulary object descriptions), b) **attribute**, c) **spatial relations**, d) **visual relations**, e) **semantic relations**, f), **multi-hop reasoning**, and g) **user-affordances**. The number of annotations per query type given in Table 1. We make sure to include unique test scenes from the dataset and include images with heavy clutter. The target of each scene within a query type is unique, and we make sure to include images with distractor objects (of the same category as the target) for all query types that require relational reasoning (all except name and affordance).

Regarding our custom FGVP-CLIP baseline (FGVP*), we present analytical comparisons and ablation in the following subsection.

## C.2 Baselines Implementation and Ablations

We utilize the provided demo applications for the end-to-end methods (SEEM, PolyFormer) to conduct grounding experiments manually. For CLIP-based baselines, we re-implement all methods from the corresponding papers (ReCLIP, RedCircle, FGVP) . We use the ViT-B visual encoder to extract features from image segments and the default BERT text encoder to represent the input query. CLIP-based baselines compute the cosine similarity between segment and text features to rank them and select the most similar segment as the final result via the argmax operator. Ground-truth masks are used for all CLIP-based baselines, similar to GPT-4v ones. We would like to highlight that in the original papers, the aforementioned baselines use potential post-processing steps to enhance the grounding capabilities of CLIP. In particular, ReCLIP uses syntactic parsing to extract entity and relation words/phrases from the input query, as well as spatial relation resolution heuristics (e.g. 'left', 'on' etc. - designed specifically for the RefCOCO dataset) to process the relations analytically and combine CLIP predictions only for the entities. RedCircle and FGVP additionally utilize a *"subtraction"* post-processing step, where they further subtract from the similarity values the average in a set of mined hard-negative queries (again selected for a specific dataset). We believe that such steps constitute domain-aware hand-crafted efforts, which even though helpful, do not represent the challenges of open-ended generalization, which is the primary focus of this work. As a result, we do not consider such post-processing steps in our baseline implementation.

**Comparisons with end-to-end approaches** The need for manual annotations to exhaust all possible language query inputs, as well as the need for manual testing via online demo applications for the considered specialist end-to-end methods (SEEM, PolyFormer) restrained us from conducting experiments in large-scale. Instead, we originally conducted experiments in a smaller subset of 52 images. Results are given in Table 3. Results follow similar patterns to the larger test set of the main paper. Specialist models (SEEM, PolyFormer) struggle with even simple name queries, scoring below 15% on average. This is potentially due to the high discrepancy between the training distribution of RefCOCO and Visual Genome and our test data, as well as the lack of relational and affordance-based language in these datasets. GPT-4v-based methods still compare favourably to CLIP-based baselines, even in the SoM setting where single marked image is used. Overall, our OWG-grounder achieves an averaged mIoU score of $70.4\%$, which is almost $\times 2$ from the previous approach.

**CLIP Visual Prompt Ablations** To further analyze the performance of CLIP-based baselines, we conduct ablation studies where we use specific elements of each method. In particular, we study: a) effect of using **multi-templates** for the text prompt, where we average text embeddings from multiple versions of the query, using templates from the original paper, b) averaging similarity scores from the visual prompt and **crops** of each segment, as originally proposed in ReCLIP, c) different visual prompt schemes, like drawing a boundary (**rectangle** or **ellipse** - as in RedCircle), converting

| w/ Crop | w/ White-Back. | w/ Blur-Rev | w/ Gray-Rev | w/ Multi Temp. | Rect. | Ellipse | Mask | mIoU |
|---|---|---|---|---|---|---|---|---|
| | | | | | | | ✗ | 18.3 |
| | | | | | | ✗ | | 31.1 |
| | | | | | ✗ | | | 34.8 |
| | | | | ✗ | ✗ | | | 33.7 |
| | | | ✗ | | ✗ | | | 24.6 |
| | | ✗ | | | ✗ | | | 26.3 |
| | | ✗ | ✗ | | ✗ | | | 34.9 |
| | | ✗ | ✗ | | ✗ | | ✗ | 41.5 |
| | | ✗ | ✗ | ✗ | ✗ | | ✗ | 43.0 |
| | ✗ | ✗ | ✗ | ✗ | ✗ | | ✗ | **51.8** |
| ✗ | ✗ | ✗ | ✗ | ✗ | ✗ | | ✗ | 51.2 |

Table 2: Component ablation studies for CLIP-based visual prompting. Results in %.

| Method | Found. Model | Name | Attribute | Spatial Relation | Visual Relation | Semantic Relation | Affordance | Multi-hop | Avg. |
|---|---|---|---|---|---|---|---|---|---|
| PolyFormer | - | 20.9 | 13.3 | 2.6 | 0.8 | 3.1 | 6.7 | 8.3 | 8.0 |
| SEEM | - | 23.3 | 10.1 | 4.6 | 10.5 | 10.2 | 7.9 | 17.5 | 12.1 |
| ReCLIP | CLIP | 36.9 | 40.0 | 12.7 | 14.2 | 20.1 | 23.0 | 34.0 | 25.9 |
| RedCircle | CLIP | 33.3 | 21.1 | 19.7 | 15.4 | 18.8 | 24.0 | 47.4 | 25.7 |
| FDVP | CLIP | 25.1 | 19.0 | 23.7 | 25.2 | 12.3 | 22.5 | 22.8 | 21.6 |
| SoM | GPT-4v | 40.1 | 25.0 | 23.3 | 40.3 | 42.5 | 60.0 | 21.2 | 36.1 |
| OWG (Ours) | GPT-4v | 83.3 | 80.1 | 45.7 | 55.4 | 78.8 | 90.3 | 59.4 | 70.4 |

Table 3: Segmentation - mIoU(%) results in different language input types for cluttered indoor scenes from OCID.

to **grayscale** or **blurring** the rest of the frame (as proposed in FGVP), as well as a prompt that we discover ourselves works good, using a **white background** for the rest of the frame. We note that in our paper's results the element combinations we used are the following:
**ReCLIP**: rectangle prompt, multi-templates, blur-reverse + crop,
**RedCircle**: ellipse prompt, multi-templates, gray-reverse + blur-reverse,
**FGVP**: mask prompt, multi-templates, gray-reverse + blur-reverse

Ablation results are shown in Table 2. Our findings are the following: 1) drawing a rectangle prompt outperforms ellipse and mask (object contours) in itself, but ensembling rectangles and masks gives the best result, 2) using multiple text templates outperforms single-template only when ensembling multiple visual inputs, c) the most effective component is our method of replacing the rest of the frame with white background, compared to grayscale and reverse operators of FGVP, while ensembling all together gives the best performance. We call our custom FGVP baseline FGVP$^*$. We present analytical results per query type for CLIP-based baselines versus GPT-4v methods, as in the original paper, for our extended evaluation dataset in Table **??**. FGVP$^*$ represents the best configuration of CLIP-based visual prompting as found by our ablation experiments. Results follow similar patterns to the smaller subset of the main paper, with a significant performance boost for CLIP-based baselines. However, GPT-4v-based methods still compare favourably to CLIP-based baselines, even in the SoM setting where single marked image is used. Our OWG visual prompt scheme dramatically outperforms all baselines, with a margin of 27.7% from SoM and 29.0% from the best found CLIP visual prompt methodology, showcasing its superiority in cluttered indoor scenes context as in OCID.

### C.3 Instance Segmentation Ablations

We use the checkpoints provided by the authors for UOIS [11] unseen object instance segmentation, as well as the ViRL-RPN checkpoint and hyper-params from the implementation in this HTTP URL. For SAM, we use the ViT-L variant of the released SAM [12] checkpoints, and search for optimal hyper-parameters for automatic mask generator, resulting in the following configuration: `points_per_side=24, pred_iou_thresh=0.92, stability_score_thresh=0.95`. We apply non-maximum suppression with an `iou_threshold=0.5` and remove nested masks, i.e. masks that are completely inside other masks of higher score threshold. This step aids in keeping only object-level SAM predictions and decreasing the over-segmentation behavior that default SAM provided in our first implementation. In turn, this leads to less cluttered visual markers for our OWG grounding module. Example instance segmentation masks for the different methods are illustrated in Fig. 5.

## D GPT-4v Example Responses

In Figs. 6, 7, 8, we provide example responses for grounding different types of language queries in OCID scenes. We observed that GPT-4v, augmented with marked image prompting, can ground not just object-related queries but also complex referring expressions that require reasoning about space, visual attributes, semantics and user-affordances. Interestingly, we find that GPT-4v responds

to queries that require symbolic reasoning concepts such as counting and negation, which are notoriously hard to emerge in specialist grounding models. In Fig. 9, we provide some example responses corresponding to failure cases. Main failure modes include: a) grounding a distractor instead of the desired object, b) not finding the object of interest at all, c) providing a correct reasoning and identifying the target in the raw image, but providing a wrong ID of an irrelevant object.

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

You are only allowed to use the provided skills. It's essential to stick to the format of these basic skills. When creating a plan, replace the object ID placeholders with specific object instances using their unique ID in square brackets as in the examples above.

Use the following two examples as a demonstration of how you should interact with the tools to control the robot and understand what the response format should be like.

### Start of episode
User Input: "It's dark around here, I need some light".
---
Observation:
'ID': 1, 'name': 'binder', 'x1y1x2y2': [0.49, 0.11, 0.68, 0.49]
'ID': 2, 'name': 'cereal box', 'x1y1x2y2': [0.2, 0.27, 0.42, 0.48]
'ID': 3, 'name': 'food bag', 'x1y1x2y2': [0.66, 0.6, 0.78, 0.81]
'ID': 4, 'name': 'food can', 'x1y1x2y2': [0.3, 0.66, 0.38, 0.78]
'ID': 5, 'name': 'cereal box', 'x1y1x2y2': [0.23, 0.09, 0.47, 0.31]
'ID': 6, 'name': 'kleenex', 'x1y1x2y2': [0.37, 0.47, 0.55, 0.61]
'ID': 7, 'name': 'flashlight', 'x1y1x2y2': [0.45, 0.79, 0.48, 0.95]
'ID': 8, 'name': 'marker', 'x1y1x2y2': [0.47, 0.73, 0.59, 0.85]
'ID': 9, 'name': 'glue stick', 'x1y1x2y2': [0.45, 0.63, 0.48, 0.7]
#Robot thought: The user mentions "It's dark around here, I need some light", indicating that they would like to use a tool that can act as a light source, such as a light bulb, a lamp or a flashlight. A flashlight exists in the scene with ID [7], so I should pick it to provide it to the user. After inspecting the location coordinates of flashlight [7] compared to other objects, it appears that there is an overlap with the marker [8]. x1=0.47 for marker [8] is between x1=0.45 and x2=0.48 of flashlight [7], and y2=0.85 for marker [8] is between y1=0.79 and y2=0.95 of flashlight [7]. This indicates that directly picking flashlight [7] could lead to a failure, so I should first remove the neighboring marker [8].
#Robot action: remove [8]
Observation: You failed to grasp object [8]
#Robot thought: I failed to grasp the marker [8], so I should try again.
#Robot action: remove [8]
Observation: You successfully removed object [8]
#Robot thought: Since the target (flashlight [7]) is now available, I should pick it to give it to the user.
#Robot action: pick [7]
Observation: You successfully picked object [7]
#Robot thought: I have completed the task.
#Robot action: done
---
### End of episode

### Start of episode
User Input: "get the corn flakes next to the gray keyboard".
---
Observation:
'ID': 1, 'name': 'cereal box', 'x1y1x2y2': [0.57, 0.28, 0.82, 0.47]
'ID': 2, 'name': 'marker', 'x1y1x2y2': [0.39, 0.44, 0.7, 0.64]
'ID': 3, 'name': 'flashlight', 'x1y1x2y2': [0.38, 0.75, 0.49, 0.85]
'ID': 4, 'name': 'cereal box', 'x1y1x2y2': [0.24, 0.46, 0.45, 0.69]
'ID': 5, 'name': 'keyboard', 'x1y1x2y2': [0.6, 0.8, 0.95, 0.97]
'ID': 6, 'name': 'keyboard', 'x1y1x2y2': [0.19, 0.56, 0.32, 0.67]
'ID': 7, 'name': 'shampoo', 'x1y1x2y2': [0.74, 0.48, 0.81, 0.6]
'ID': 8, 'name': 'unknown', 'x1y1x2y2': [0.62, 0.64, 0.71, 0.74]
'ID': 9, 'name': 'kleenex', 'x1y1x2y2': [0.46, 0.2, 0.56, 0.41]
'ID': 10, 'name': 'soda can', 'x1y1x2y2': [0.32, 0.69, 0.38, 0.71]
'ID': 11, 'name': 'ball', 'x1y1x2y2': [0.55, 0.57, 0.63, 0.69]
'ID': 12, 'name': 'hand towel', 'x1y1x2y2': [0.17, 0.71, 0.35, 0.95]
'ID': 13, 'name': 'stapler', 'x1y1x2y2': [0.53, 0.7, 0.57, 0.84]
'ID': 14, 'name': 'ball', 'x1y1x2y2': [0.6, 0.7, 0.67, 0.79]
'ID': 15, 'name': 'food bag', 'x1y1x2y2': [0.17, 0.41, 0.28, 0.58]
'ID': 16, 'name': 'food can', 'x1y1x2y2': [0.65, 0.32, 0.71, 0.4]
'ID': 17, 'name': 'sponge', 'x1y1x2y2': [0.35, 0.59, 0.42, 0.7]
#Robot thought: The user mentions "get the corn flakes next to the gray keyboard", so I should: 1) identify the gray keyboard, 2) identify the corn flake objects, 3) filter the one whose next to the gray keyboard, 4) pick the filtered corn flakes. There are two keyboard items in the scene ([5], [6]), I should find which one is the gray one. I will invoke the visual grounding tool to determine which one of the two keyboards is gray.
#Robot action: visual_grounding [5,6] `gray'
Observation: Keyboard [6] is gray.
#Robot thought: I identified the gray keyboard [6]. Now I have to identify the corn flake objects. There are two cereal box items in the scene ([1], [4]). I have to look at their location coordinates and identify which one is next to the keyboard.
Keyboard [6] coordinates are: (x1=0.19, y1=0.56, x2=0.55,y2=0.76), so the center would be: (x0=0.255, y0=0.66)
Cereal box [1] coordinates are: (x1=0.57, y1=0.28, x2=0.82, y2=0.47), so the center would be: (x0=0.695, y0=0.375)
Cereal box [4] coordinates are: (x1=0.24, y1=0.46, x2=0.45, y2=0.69), so the center would be: (x0=0.345, y0=0.575)
Therefore, the cereal box that is closest to the keyboard [6] is cereal box [4]. After inspecting the location coordinates of cereal box [4] compared to other objects, it appears that there is an overlap with four other objects:
Marker [2] coordinates are: (x1=0.39, y1=0.44, x2=0.7,y2=0.64) which are overlapping with cereal box [4] coordinates: (x1=0.24, y1=0.46, x2=0.45, y2=0.69).
Keyboard [6] coordinates are: (x1=0.19, y1=0.56, x2=0.55,y2=0.76) which are overlapping with cereal box [4] coordinates: (x1=0.24, y1=0.46, x2=0.45, y2=0.69).
Soda can [10] coordinates are: (x1=0.32, y1=0.69, x2=0.38,y2=0.71) which are overlapping with cereal box [4] coordinates: (x1=0.24, y1=0.46, x2=0.45, y2=0.69).
Sponge [17] coordinates are: (x1=0.35, y1=0.59, x2=0.42,y2=0.7) which are overlapping with cereal box [4] coordinates: (x1=0.24, y1=0.46, x2=0.45, y2=0.69).
This indicates that directly picking cereal box [4] could lead to a failure, so I should first remove the neighbouring objects ([2], [6], [10], [17]).
#Robot action: remove [2]
Observation: You successfully removed object [2]
#Robot thought: The marker [2] is out of the way. I will proceed with removing the keyboard [6].
#Robot action: remove [6]
Observation: You failed to grasp object [6]
#Robot thought: I failed to grasp the keyboard [6], so I should try again.
#Robot action: remove [6]
Observation: You failed to grasp object [6]
#Robot thought: I failed to grasp the keyboard [6], so I should try again.
#Robot action: remove [6]
Observation: You successfully removed object [6]
#Robot thought: The marker [2] and the keyboard [6] are out of the way. I will proceed with removing the soda can [10].
#Robot action: remove [10]
Observation: You successfully picked object [10]
#Robot thought: The marker [2], the keyboard [6] and the soda can [10] are out of the way. I will proceed with removing the sponge [17].
#Robot action: remove [17]
Observation: You failed to grasp object [17]
#Robot thought: I failed to grasp the sponge [17], so I should try again.
#Robot action: remove [17]
Observation: You successfully picked object [17]
#Robot thought: The marker [2], the keyboard [6], the soda can [10] and the sponge [17] are out of the way. Since the target (cereal box [4]) is now available, I should pick it to give it to the user.
#Robot action: pick [4]
Observation: You successfully picked object [4]
#Robot thought: I have completed the task.
#Robot action: done
---
### End of episode

Now begin:
### Start of episode
User Input: "{user_input}".
---
{episode_history}
Observation: {obs}
#Robot thought:"""

Figure 2: System prompt and in-context examples for the LLM in the SayCan-IM baseline. The input field `user_input` corresponds to the task given by the user, `obs` contains the scene state or feedback from actions that is updated in each episode step, and `episode_history` the previous observation-thought-action cycles in the current episode.

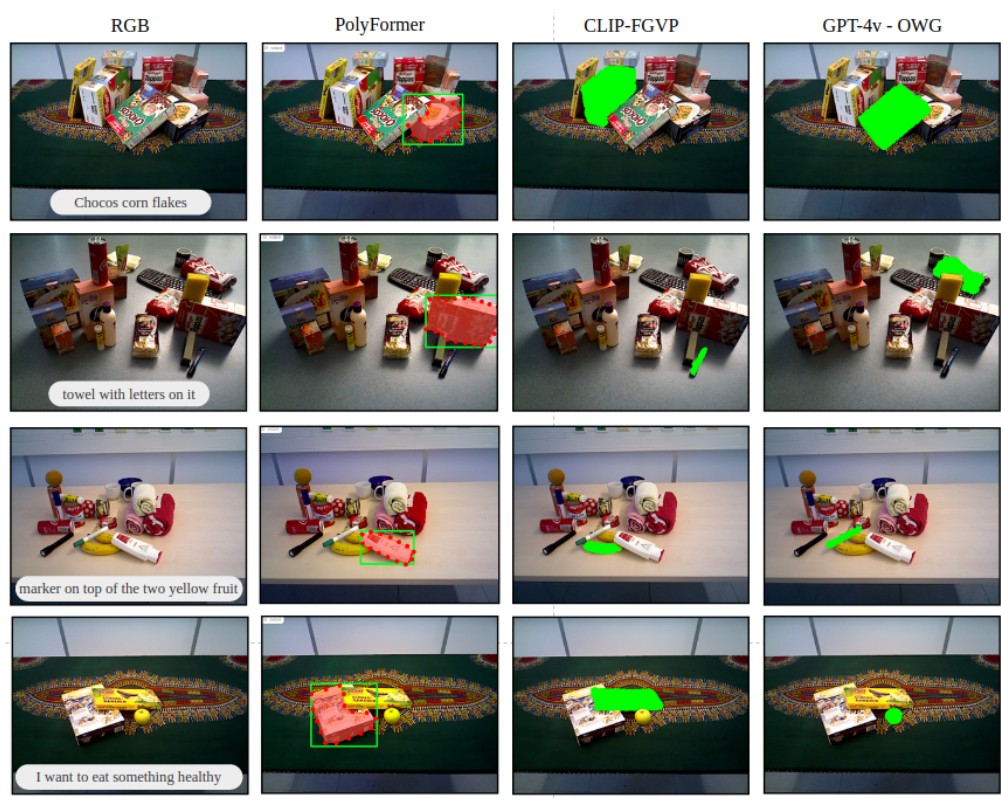

Figure 3: Example output segmentation masks of different grounding methods for OCID scenes.

OCID Segmentation Mask comparisons

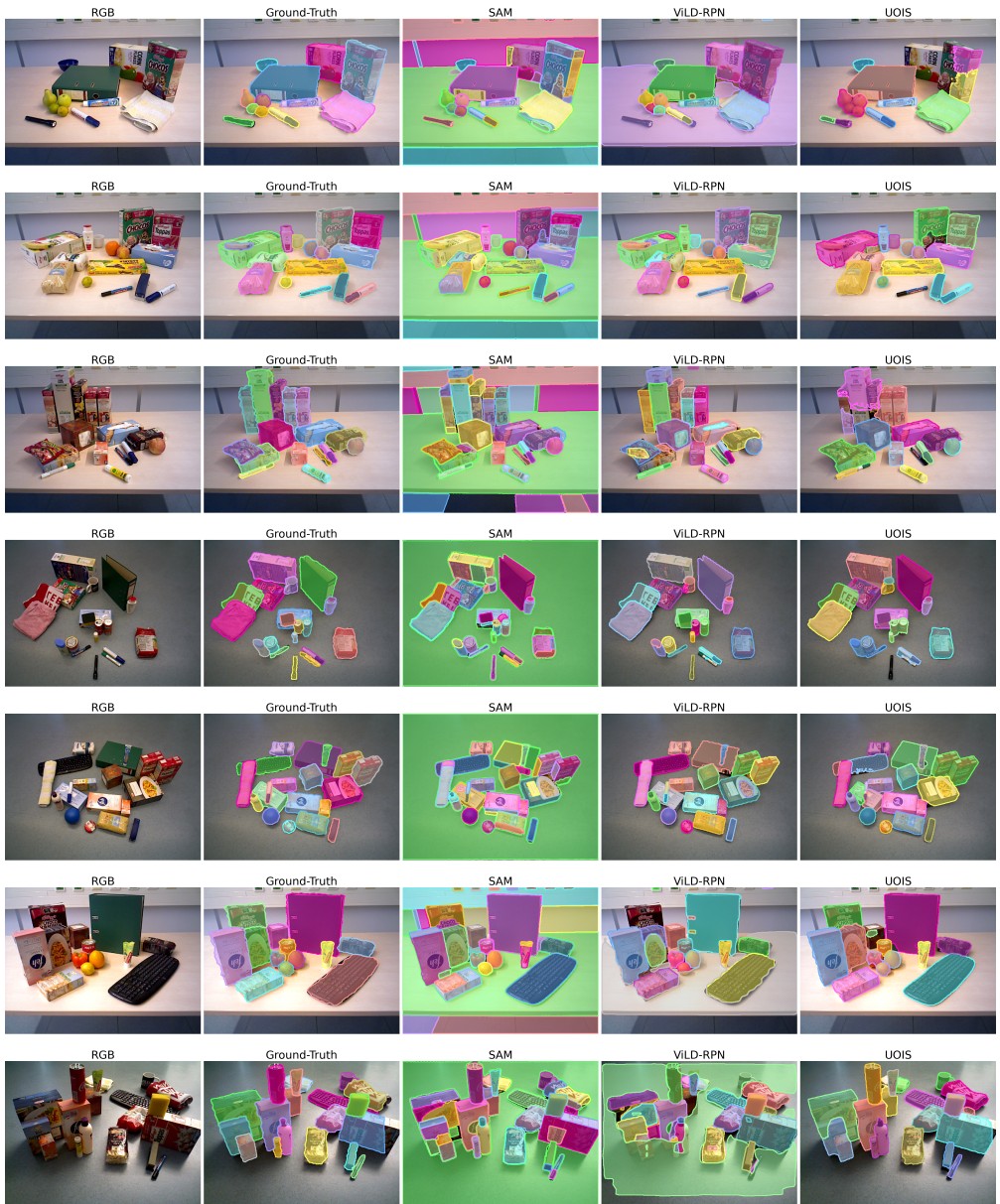

Figure 4: Comparison of different segmentation models for seven example cluttered indoor scenes from OCID dataset

OCID Segmentation Mask comparisons

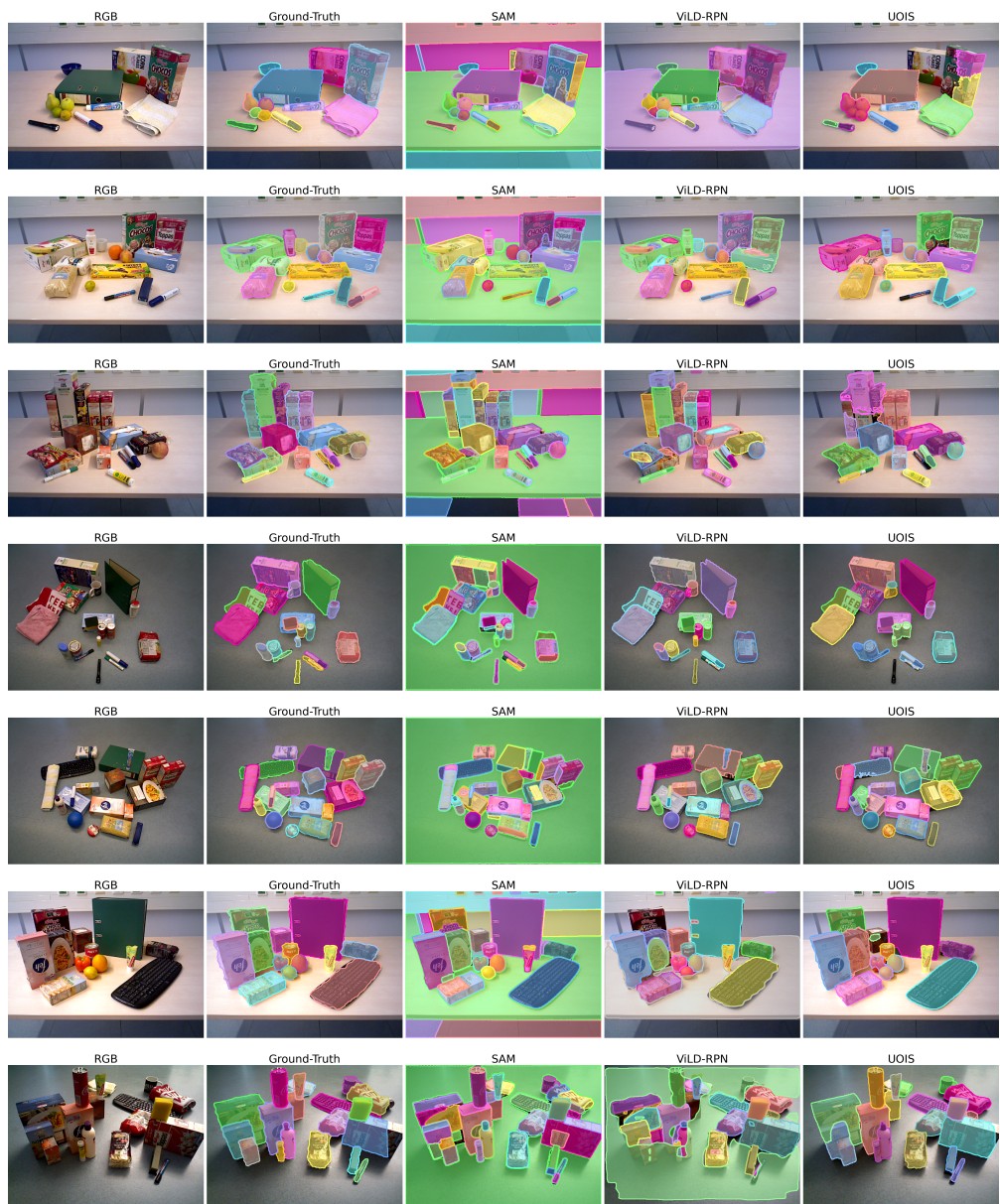

Figure 5: Comparison of different segmentation models for seven example cluttered indoor scenes from OCID dataset

## Object Brand Name

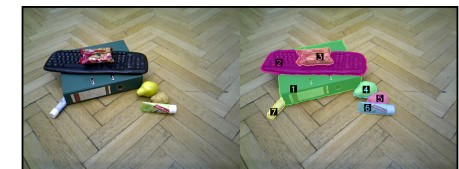

**User**: Feh package

**GPT-4v**: To find the object that matches the description "Feh package," I will first look at the raw image to identify any packages that could potentially match the description. Upon examining the raw image, I notice a package with the text "Feh" on it. This package is located towards the left side of the image, standing upright, and appears to be a box of some sort. Next, I will look at the marked image to find the same package and determine its unique ID. The package with the text "Feh" on it in the raw image corresponds to the package in the marked image with the ID [12] on it. The package is in the same position in both images, confirming that this is the correct object.
My final answer is: [12].

## Object Attribute: Maturity

**User**: non ripe pear

**GPT-4v**: To find the 'non ripe pear,' I will first identify the pears in the raw image. There are two pears visible on the floor. To determine which one is not ripe, I will look for visual cues such as color. Typically, a non-ripe pear may have a greener and firmer appearance compared to a ripe one, which would be more yellow and possibly softer looking. In the raw image, the pear on the left appears greener and is likely the non-ripe pear, while the pear on the right seems more yellow and could be considered ripe. Now, I will look at the marked image to find the ID corresponding to the greener pear. In the marked image, the greener pear on the left has the ID [4]. Therefore, this is the ID that corresponds to the 'non ripe pear.'
My final answer is: [4]

## Object Content Type

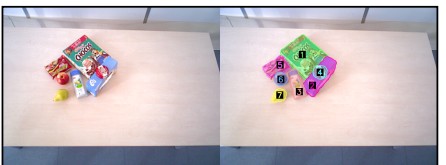

**User**: noodles

**GPT-4v**: To find the object that matches the description "noodles," I will first examine the raw image to identify any items that could be noodles. Upon inspection, I see a package with an image that appears to represent noodles on the left side of the table. Next, I will look at the marked image to find the same package and determine its unique ID. The package with the noodles is highlighted with a unique color and is associated with the ID [5].
My final answer is: [5].

## Object Attribute: Colors

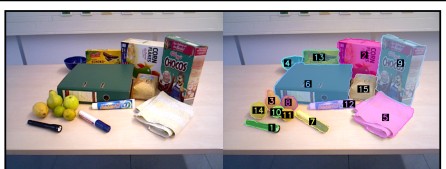

**User**: green and red corn flakes

**GPT-4v**: To find the "green and red corn flakes," I will first look at the raw image to identify the box of corn flakes. In the image, there are two cereal boxes visible. One of them has a predominantly green color scheme with some red elements, which matches the description of "green and red corn flakes." This box is on the right side of the image, next to another cereal box with a blue and yellow color scheme. Now, I will switch to the marked image to find the same box of corn flakes with the corresponding ID. The box with the green and red color scheme is highlighted with the same color as the ID number 9.
My final answer is: [9].

## Object Functionality

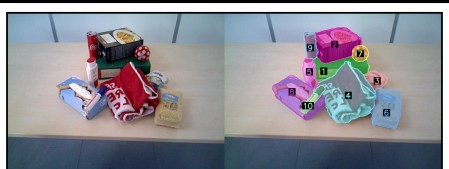

**User**: red soft thing for the hair

**GPT-4v**: To find the object that matches the description "red soft thing for the hair," I will follow these steps: 1. Identify any red objects in the raw image, 2. Determine which of these red objects could be described as "soft" and used for the hair. In the raw image, there is a red object with white text that appears to be a towel, which is typically soft and can be used for drying hair. This object is located in the center of the image, draped over other items. Now, I will look at the marked image to find the same object and its corresponding ID. The red towel in the marked image has been highlighted with a unique color and is labeled with the ID number 4.
My final answer is: [4].

## Object Attribute: Shape

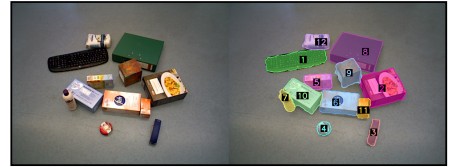

**User**: cube

**GPT-4v**: To find the object that matches the description "cube," I will first look at the raw image to identify any objects that have a cubic shape. In the raw image, there is an object that appears to be a cube towards the right side of the image, slightly above the center. It has a rusted appearance and seems to be a cube due to its equal-length edges and square faces. Now, I will look at the marked image to find the same object with its corresponding ID. The object with the cubic shape in the same location is highlighted with a color and has the ID number 9 next to it.
My final answer is: [9]

Figure 6: Example GPT-4v responses (1/3): object-related queries

## Relations: Relative Location w.r.t Object

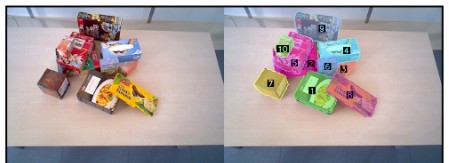

**User**: food box under the choco-banana

**GPT-4v**: To find the "food box under the choco-banana," I will first identify the choco-banana in the raw image. The choco-banana appears to be a box with an image of a banana and chocolate on it, located towards the bottom right of the image. Now, I will look for any food box that is positioned under it. In the raw image, there is a box directly underneath the choco-banana box. This box is likely the "food box" mentioned in the description. To find the corresponding ID, I will now switch to the marked image and look for the same box under the choco-banana box, which is labeled with ID 8. Directly underneath the choco-banana box with ID 8, there is another box highlighted with a different color and labeled with ID 1. This is the food box that is under the choco-banana.
My final answer is: [1].

## Relations: Absolute Location

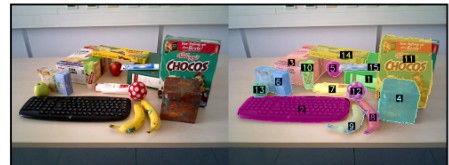

**User**: leftmost toothpaste product

**GPT-4v**: To find the "leftmost toothpaste product," I will first identify any toothpaste products in the raw image. Looking at the raw image, I can see two items that resemble toothpaste tubes. They are located towards the center of the image, one standing upright and the other lying down to its right. Now, I will switch to the marked image to find the corresponding IDs for these toothpaste products. The leftmost toothpaste product, which is standing upright, has a highlighted color and an ID next to it. The ID next to this toothpaste product is [10].
My final answer is: [10].

## Relations: Same Attribute

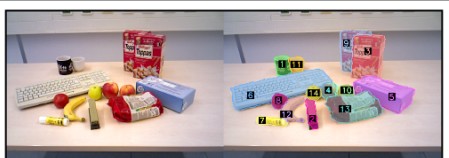

**User**: mug of same color as the keyboard

**GPT-4v**: To find the mug of the same color as the keyboard, I will first identify the color of the keyboard in the raw image. The keyboard appears to be white. Next, I will look for a mug that is also white. In the raw image, there are two mugs on the table to the left of the keyboard. One of these mugs is white, which matches the color of the keyboard. Now, I will locate this white mug in the marked image to find its corresponding ID. In the marked image, the white mug is highlighted with a unique color and has the ID number next to it. The white mug has the ID [11] next to it.
My final answer is: [11].

## Relations: Superlative Attribute

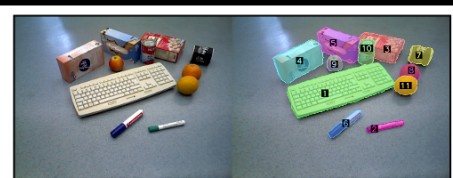

**User**: thickest marker

**GPT-4v**: To find the "thickest marker," I will first identify all the markers in the raw image. There are two markers visible: one with a blue cap and one with a green cap. Between these two, the marker with the blue cap appears to be thicker than the one with the green cap. Now, I will look at the marked image to find the ID corresponding to the thicker marker with the blue cap. In the marked image, the thicker marker with the blue cap is highlighted and has the ID number next to it. The thicker marker with the blue cap is associated with the ID [6].
My final answer is: [6].

## Multi-Hop Reasoning: Counting

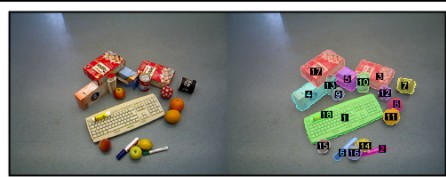

**User**: fourth spherical object from the top

**GPT-4v**: To find the "fourth spherical object from the top" in the raw image, I will first identify all the spherical objects and then count from the top of the image down to the fourth one. In the raw image, starting from the top, I see several spherical objects which appear to be fruits and a ball. Counting these objects from the top, the first spherical object is an orange, the second is another orange, the third is the red ball, and the fourth is yet another orange. Now, I will look at the marked image to find the ID corresponding to the fourth spherical object from the top, which is the orange. In the marked image, this orange is highlighted with a color and has the ID number next to it. Upon examining the marked image, I can see that the fourth spherical object from the top, the orange, has the ID number 11 next to it.
My final answer is: [11].

## Multi-Hop Reasoning: Negation

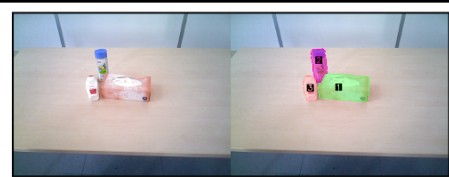

**User**: shampoo that is not standing upright

**GPT-4v**: To find the shampoo that is not standing upright, I will first look at the raw image to identify any objects that resemble shampoo bottles. In the raw image, there are two bottles on the table. One is standing upright, and the other is lying on its side. Now, I will look at the marked image to find the ID of the shampoo bottle that is not standing upright. The bottle lying on its side has a unique color and a numeric ID next to it. This ID is [3].
My final answer is: [3].

Figure 7: Example GPT-4v responses (2/3): relational reasoning queries

## Semantic Reasoning: Utility

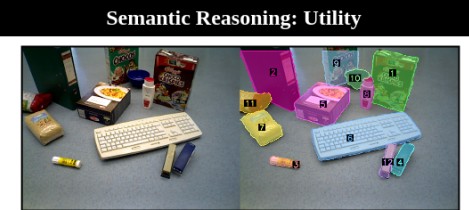

**User**: something to put my cereal in

**GPT-4v**: To find "something to put my cereal in," I'm looking for a bowl or a similar container that would be appropriate for holding cereal. In the raw image, I can see a bowl on the left side of the table, near the center of the image. It's placed in front of a cereal box and to the left of a keyboard. Now, I'll look at the marked image to find the same bowl and its corresponding ID. The bowl is in the same location as in the raw image, and it has a unique color and a numeric ID next to it. The ID next to the bowl is [10].

My final answer is: [10].

## Semantic Reasoning: Material

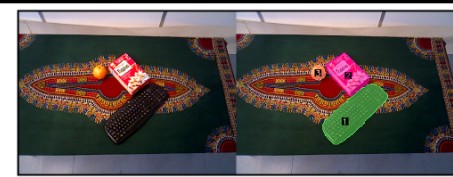

**User**: something that grows organically

**GPT-4v**: In the raw image, there is an orange, which is an object that grows organically. Now, I will look at the marked image to find the same orange and its corresponding ID. The orange in the marked image is highlighted with a unique color and has the numeric ID 3 next to it.

My final answer is: [3].

## User Affordance: State

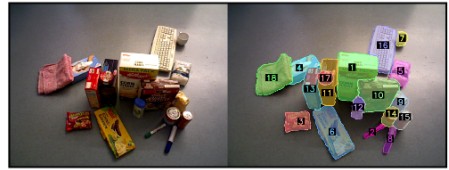

**User**: my hands are wet

**GPT-4v**: To find the object that best matches the description "My hands are wet," I need to identify an object that could be associated with wet hands. In the context of the items present in the image, the most likely candidate would be a towel or a cloth that one would use to dry their hands. Looking at the raw image, I can see a pink towel-like object on the left side of the image. This object seems to be the one that would be used to dry wet hands. Now, I will look at the marked image to find the same towel-like object and its corresponding ID. The pink towel-like object in the marked image has the ID number highlighted in the same color as the object's mask. The ID number for the towel-like object is [18].

My final answer is: [18].

## User Affordance: Function

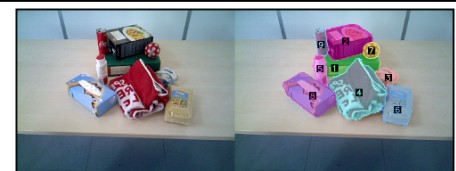

**User**: I need a toy to release some stress

**GPT-4v**: To find a toy to release some stress, I will look for an object that is commonly used for stress relief, such as a stress ball or a fidget toy. In the raw image, I can see an object that resembles a ball with a pattern on it, which could potentially be a toy used for stress relief. It is located towards the right side of the image, near the center, and is red with white spots. Now, I will look at the marked image to find the same object and its corresponding ID. The object that matches the description and location in the raw image is highlighted with a unique color and has the ID number next to it. In the marked image, the ball with the red and white pattern has been highlighted and is associated with the ID number [7].

My final answer is: [7].

Figure 8: Example GPT-4v responses (3/3): semantics-related queries

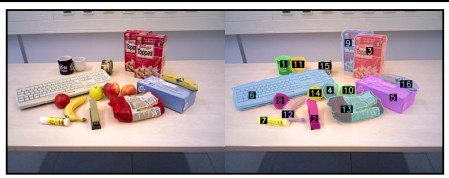

**User**: canned food

**GPT-4v**: To find the "canned food" in the raw image, I will look for an object that resembles a can or a container typically used for preserving food. In the raw image, I can see a box of what appears to be cereal on the right side, but that is not canned food. However, there is no object that clearly resembles a traditional metal can used for canned food. Since there is no object that matches the description of "canned food" in the raw image, I cannot provide an ID from the marked image. Therefore, there is no label ID to match the description of "canned food" in these images.

My final answer is: there is no label ID for "canned food" in the provided images.

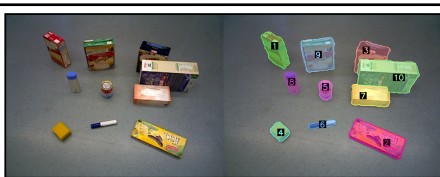

**User**: corn flakes that is laying on its side

**GPT-4v**: To find the "corn flakes box that is laying on its side," I will first look at the raw image to identify the box that resembles a corn flakes box and is positioned on its side. In the raw image, there is a box with what appears to be a corn flakes design on the bottom right, and it is indeed laying on its side. Next, I will look at the marked image to find the same box and determine its unique ID. The box that matches the description in the marked image is highlighted with a color and has the number 2 next to it.

My final answer is: [2].

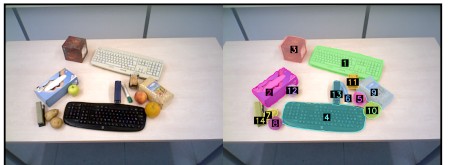

**User**: stapler next to the marker

**GPT-4v**: To find the stapler standing next to the marker, I will first identify these objects in the raw image and then locate their corresponding IDs in the marked image. In the raw image, I can see a stapler to the left of a black keyboard, and immediately to the right of the stapler is a marker. Now, I will switch to the marked image to find the IDs that correspond to these objects. In the marked image, the stapler has a unique color and ID next to it, and the marker also has a different color and ID. The stapler is marked with the ID [7], and the marker is right next to it with the ID [6].

My final answer is: [7].

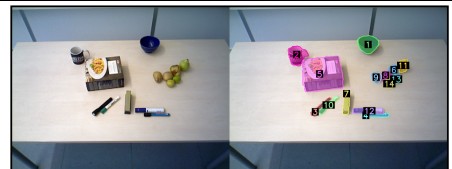

**User**: blue and black marker

**GPT-4v**: ITo find the "blue and black marker," I will first look at the raw image to identify any markers that match this description. In the raw image, there are several markers on the table. I can see one marker that appears to have a blue cap and a black body, which is located towards the bottom right of the table. Now, I will switch to the marked image to find the same marker with the corresponding ID. The marker with the blue cap and black body in the marked image has a unique color and ID number next to it. It is the same marker that I identified in the raw image, and it has the ID number [12] next to it.

My final answer is: [12].

Figure 9: Example of GPT-4v response failures.

