# OpenReview forum: "Towards Open-World Grasping with Large Vision-Language Models"
_robot-learning.org/CoRL/2024/Conference — CoRL 2024_

### Official Review · Reviewer_PGH6 · 2024-07-19

**Originality:** 3
**Technical Quality:** 2
**Clarity Of Presentation:** 2
**Potential Impact:** 3
**Recommendation:** 3
**Confidence:** 4

**Review:**

## Strengths

Using VLM is essential for robots to effectively perform open-world grasping under diverse cluttered scenes in the real-world that potentially require the robot to (1) plan and perform multiple actions (e.g., removing obstacles) before grasping the target object, and (2) handle arbitrary unstructured user instructions in daily life. The authors investigate and explore different design decisions that effectively integrate VLM into the open-world grasping pipeline, providing meaningful insights for the community.


## Weaknesses and Questions


### Methodology

- The method to use self-consistency prompting and to use high-resolution image input (Table 3) are not defined in the paper, even though they provide significant benefits to the proposed approach's object-grounding performance.
- Is using VLM for grasp ranking really helpful for performance, and how does it compare with using the grasp rankings from off-the-shelf grasping models? (Minor: With a stronger grasping model than the one used in the paper (e.g., Contact-GraspNet, Anygrasp), does it make a difference?)


### Experiments

- According to Table 2, the real-world evaluation is only conducted for 6 trials for both seen / unseen instruction settings. The results can be sensitive to the noise resulting from a low number of evaluation trials. Increasing the real-world evaluation to at least 12 trials per setting will make the results more compelling.
- For Table 4, authors only conducted a small number of trials (15) for simuation experiments, and the results are quite sensitive to noise. One should conduct a much larger number of simulation evaluation trials (e.g., 50 / 100), especially since simulation evaluations are more scalable than real evaluations.
- For Table 1, is the comparison with baselines fair, i.e., the baselines are also using ground-truth segmentation masks to calculate mIoU?
- The ablation experiment using SAM for instance segmentation reports surprising low numbers, which seems to be a mistake. One should use the lowest granularity mask in SAM or use the "return single-mask" mode in SAM.



### Writing

- The introduction and motivation of the work are weak, tedious, and do not reveal the true significance of VLMs in a robot grasping pipeline. The main significance of VLMs in robot grasping comes from their ability to handle complex cluttered scenarios that potentially require multi-step reasonings & robot actions, and to handle arbitrary, unstructured user instructions encountered in the daily world. In particular, previous works primarily focus on simple scenarios where the robot can directly grasp the target object upon seeing it. However, in more complex, cluttered settings typically encountered in the real world, some objects may obstruct the target object, and they need to be moved away before the robot can grasp the target object. In these scenarios, VLMs become essential for accomplishing the task, which is the focus of the paper. On the other hand, VLMs are less beneficial in simple grasping scenarios, where one can directly employ open-world detection-segmentation pipeline (e.g., GroundingDINO + SAM) and existing grasping models (e.g., AnyGrasp) to accomplish the task.

**Quality Of The Limitations Section:**

3

**Questions For Rebuttal:**

Please see "Weaknesses and Questions" in the review and address the raised issues in methodology, experiments, and writing.

**Robotics Focus:**

4

**Summary Of Paper:**

This paper proposes a method that integrates vision language models (e.g., GPT-4o) into the open-world robot grasping pipeline. Through the use of VLM, the grasping agent is capable of handling arbitrary user instructions and also effective under cluttered grasping scenarios. Specifically, the authors propose to use set-of-marks prompting and chain-of-thought prompting to generate grasping plans, especially in clutter scenarios where some objects might need to be removed first before the robot can grasp the target object. The authors also propose to use VLM to rank grasping proposals for better grasping qualities.

**Summary Of Recommendation:**

This paper provides some meaningful contributions on integrating VLM into the open-world grasping pipeline, but suffers from some notable weaknesses in methodology, experiments and writing. Edit: Most of them are addressed by authors' rebuttal, so I raised my score.

---

### Official Review · Reviewer_hpr3 · 2024-07-24

**Originality:** 2
**Technical Quality:** 2
**Clarity Of Presentation:** 3
**Potential Impact:** 2
**Recommendation:** 2
**Confidence:** 4

**Review:**

**Strengths:**
1. **Timely Topic:** The integration of vision-language models (VLMs) in robotics, particularly for grasping tasks, addresses a highly relevant and current area of research.
2. **Systematic Approach:** The paper is well-organized, detailing the OWG system's methodology, which includes segmentation, grasp planning, and ranking.
3. **Comprehensive Evaluation:** Extensive testing in both simulation and real-world settings provides a thorough evaluation of the system's effectiveness.

**Weaknesses:**
1. **Limited Novelty:** The approach closely follows existing methodologies and focuses on simple pick-and-place tasks in cluttered indoor scenes, a well-explored area. The paper lacks groundbreaking innovations.
2. **Scope of Application:** The focus on common scenarios does not sufficiently demonstrate the system's applicability to more complex or diverse environments.
3. **Clarity and Detail:** While generally clear, the paper could benefit from more detailed explanations of the visual prompting mechanisms and the specifics of the ablation studies.
4. **Discussion on Limitations:** There is an insufficient critical analysis of the system's limitations, particularly regarding its generalizability and potential shortcomings in more varied scenarios.

Overall, while the paper contributes to the understanding of using VLMs for robotic grasping, it may not meet the high standards expected at CoRL due to its limited novelty and the well-trodden nature of its experiments. Future work should focus on more innovative approaches and diverse applications.

**Quality Of The Limitations Section:**

3

**Questions For Rebuttal:**

See above

**Robotics Focus:**

3

**Summary Of Paper:**

The paper presents Open World Grasper (OWG), a system that leverages modern vision-language models (VLMs) to enable robotic grasping in open-world scenarios based on open-ended language instructions. The system addresses two major limitations of previous approaches: the need for better visual grounding and the inclusion of low-level spatial understanding, which are critical for handling contact-rich environments. OWG combines VLMs with segmentation and grasp synthesis models to perform three main tasks: open-ended referring segmentation, grounded grasp planning, and grasp ranking through contact reasoning. This pipeline allows for zero-shot application, utilizing suitable visual prompting mechanisms to integrate high-level semantic reasoning with low-level physical-geometric insights. Extensive evaluations demonstrate OWG's robustness and superior performance compared to prior methods in both simulated and real-world robotic grasping experiments.

**Summary Of Recommendation:**

This paper doesn't have unique contributions and is very similar to lots of recent papers in this field, thus I would like to reject this paper.

---

### Official Review · Reviewer_Adjp · 2024-07-28
**Reflection on Zero-shot OWG using Foundation Models**

**Originality:** 3
**Technical Quality:** 3
**Clarity Of Presentation:** 4
**Potential Impact:** 3
**Recommendation:** 4
**Confidence:** 4

**Review:**

## Strengths

- **Zero-shot Performance:** The framework achieves impressive zero-shot capabilities in open-world grasping without requiring retraining or fine-tuning compared to prior methods whether supervised (CROG) or zero-shot (SayCan-IM).
- **Compositional Design:** The modular approach allows for better interpretability and analysis of errors at each stage.
- **Ablations, Benchmarking and Evaluation:** Evaluations are properly conducted, particularly evident in the diverse range of expressions used, such as attributes, spatial relations, and visual relations, to assess the initial stage in the framework for target referral. Additionally, the real-world experiments strengthen the position of the full framework especially in cluttered scenes compared to other methods.

- **Failure Recovery Mechanism:** The framework's grasp planner helps recover from failures and straightforwardly enhances the utility.

## Weaknesses

### Clarity Issues
- **Explanation of SayCan-IM Baseline:** The explanation of the SayCan-IM baseline, which contrasts the use of pure language models (LLMs) with external perception modules against the proposed framework's vision-language model (VLM) in open-world grasping, lacks clarity. The supplementary material could benefit from clearer explanations,  in-context prompts examples and additional references.
- **Appendix E Reference:** The paper references Appendix E for in-context examples, but this section does not appear in the supplementary material (Section A1.1).
Could the authors clarify whether this is a misreference or if the content is missing?

 - **Self-Consistency:** The concept of self-consistency is mentioned in the main paper without a clear reference to an appropriate section in the appendix. Could the authors specify where this discussion can be found or provide a brief explanation?
 - **Duplicate Content:** There are two nearly identical paragraphs discussing in-context examples in the supplementary material (Section A1.1). Is this redundancy intentional, or should these sections be revised for clarity?
- **OWM 2D instance segmentation model:** The authors perform an ablation study on the OWG grounder performance on a curated test set from OCID dataset. The results show that the best 2D instance segmentation model is the UOIS (Xie et. al 2021), but still the main figure mentions SAM despite poor performance. For real-world experiments, authors are clear on the 2D instance segmentation model (Mask-RCNN) as stated "261 In both
SayCan-IM and our method, Mask-RCNN [65]
is utilized for 2D instance segmentation... "
  * Is this the 2D instance segmentation model (Mask-RCNN) for the grounding experiments too?
  * Does this explain the discrepancy between the results in Table 1 and Table 3? I cannot see a matching mIOU performance between the two tables. More clarity on this issue is important.

### Related Work and Comparison
- **Task-oriented grasping methods:** The scope of this work is closely related to the task-oriented grasping field (Tang et al., 2023). However, the authors did not explicitly position their work concerning similar studies. Mentioning such work can strengthen the paper and comparing it can consolidate it more.
  * Could the authors elaborate on why there is no direct comparison with the Grasp-GPT approach?
  * What were the considerations behind not including this work in the related work section?


[1]: Tang, C., Huang, D., Ge, W., Liu, W., & Zhang, H. (2023). Graspgpt: Leveraging semantic knowledge from a large language model for task-oriented grasping. *IEEE Robotics and Automation Letters*.
[2] Xie, C., Xiang, Y., Mousavian, A., & Fox, D. (2021). Unseen object instance segmentation for robotic environments. IEEE Transactions on Robotics, 37(5), 1343-1359.

**Quality Of The Limitations Section:**

3

**Questions For Rebuttal:**

### SayCan-IM Baseline

  - What specific role does the large language model (LLM) play in the SayCan-IM baseline? Is its function limited to planning according to a predefined objective (distance to the target object center) and invoking CLIP, or does it have broader applications within the baseline?
  - Given the context of open-ended language grounding, do the authors see the potential for a more nuanced use of pure LLMs to provide? how might this offer a fairer comparison against vision-language models (VLMs)? My main concern is that there is overlapping functionality between the instance segmentation model and the VLM that can incur extra computational costs and using pure LLMs efficiently might circumvent the functional redundancy.

### Real-World Experiment Reproducibility
- **Replication Guarantees:** The setting of the real-world experiments appears difficult to replicate. Is there a mechanism, maybe the readily available instance segmentation mechanism that overlays the current scene masks and the prespecified ones and ensures consistency across all methods tested? This would help validate the experimental results and conclusions drawn.

**Robotics Focus:**

4

**Summary Of Paper:**

The work presents a simple yet novel framework which enables zero-shot open-world grasping in a three-stage setting. The first stage involves open-world object referral using open-ended language queries, enabling the identification of target objects even with relational expressions to the target. The second stage focuses on planning a sequence of actions to clear obstructing objects, ensuring a clear path for grasping the target. In the third stage, the system ranks and selects the most promising grasp from multiple proposals. The framework's effectiveness is demonstrated both quantitatively and qualitatively, particularly in cluttered environments and semantic language queries beyond the object name, outperforming previous supervised and LLM-based methods in simulation and real-world experiments.

**Summary Of Recommendation:**

The work presents promising advancements in open-world grasping for robotics as demonstrated by its substaintial improvements over supervised and LLM based methods while avoiding the retraining overhead. However, to strengthen the paper, improvements in presentation and a more comprehensive inclusion of relevant literature are recommended. I was a bit torn on the recommendation and I am relying on the authors answers to maintain it or even improve it.

---

### Official Review · Reviewer_HEJP · 2024-08-02
**Experiments section needs minor revision**

**Originality:** 3
**Technical Quality:** 4
**Clarity Of Presentation:** 3
**Potential Impact:** 3
**Recommendation:** 3
**Confidence:** 4

**Review:**

# Clarity

Most sections of the paper are well written. Following remarks might help improve the paper's presentation further:
- Notation consistency: In abstract and related works, LVLMs and VLMs are treated differently. For example, CLIP is treated as VLM, while GPT-4v is treated as an LVLM. In the method section, the authors were supposed to use the term "LVLM" to continue this consistency. However, the term "VLM" has been used to indicate an "LVLM".
- Line number 189: It must be $\hat{n}$ not $n^*$
- In the experiments section (section 4.1 and 4.2), main content did not refer to the tables present in the paper. For example, when starting the paragraph at line number 219, it would be nice to start with saying that the results for this section are plotted on table 1, and then continue discussing about various points that depend on this table. Same for section 4.2, where table 2 needs to be referenced.
- In section 4.1, the term "OWG-Grounder" is mentioned for the first time. It would be good to introduce the term while discussing the method in section 3.2 "Open Ended Referring Segmentation" paragraph.
- Ambiguity in the term "scenario". Although the video clearly explains it, It would good to mention what a "scenario" is, at the beginning of the experiments section, at least once.
- Section 4.2 "Results": In line number 288, "grasping and grasping", you may wanted to say "grounding and grasping"
- Providing a brief introduction to the OCID-VLG dataset in section 4.1 would be a good idea.
- In section 4.1, line 226, the statement " We identify two basic failure modes: a) the LVLM confused the target description with another object, e.g. due to same appearance or semantics, and b) the LVLM reasons correctly about the object and where it is roughly located, but chooses a wrong numeric ID to refer to it." sounds speculative, as it is written as a standalone statement with no reference to specific experimental data or a citation to a prior work. If you are presenting it as a qualitative observation, it would be nice to add a phrase that it is a qualitative observation.

# Quality
The quality of the paper is good. Abstract, Introduction, Related Works, and Method sections are well presented. Experiments section may need minor revision, in terms of table references, brief introduction to datasets, baselines and their relevance.

# Originality
The presented work brings together known approaches, and presents improved prompting method to a different problem. Along the way, ablation studies shown in the paper help the community understand the implications of various components in the pipeline and how they affect the overall performance.

# Relevance
This paper is quite relevant to grasping, and LVLM (Large Vision Language Models) space, showcasing:
- How to tightly integrate LVLM for grasping and bring in closed-loop framework for improved performance on cluttered scenes.
- Some good insights from pure-vision and VLM papers and their application to robotics.

# Limitations
Limitations section is well covered.

**Quality Of The Limitations Section:**

3

**Questions For Rebuttal:**

I do not have any questions after thoroughly reviewing the paper, supplementary material, the video, and some of the related works. However, the paper's main material alone is likely to raise certain ambiguities and questions. I have shared my remarks on how these issues may be reduced in the 'Review' section above.

**Robotics Focus:**

4

**Summary Of Paper:**

The authors propose a framework that utilises VLMs, more specifically LVLMs, to tackle the problem of "robotic grasping" in a setting where task instructions are open-ended language phrases. The paper argues that while LLMs are quite good at task-level abstract reasoning, they are weak at spatial and geometric reasoning, due to lack of direct access to visual information. Frameworks that propose to use VLMs such as CLiP or SAM in conjunction with LLMs, often suffer from bottlenecks presented by VLMs. LVLMs, on the other hand, are inherently trained on both vision and language data, and are hence very good at combining abstract task-level reasoning with spatial reasoning. Specifically, the framework consists of three stages, namely: "open-ended referring segmentation", "Grounded grasp planning", and "Grasp ranking via contact reasoning". In stage 1, building on prior work, the given input image is semantically grounded through labelling each object with a unque number overlayed on it. In stage 2, both the initial image, the grounded image, and the input text phrase are provided to the LVLM to produce a task plan as a sequence of objects that need to be removed or picked, in order to accomplish picking up the goal object. In stage 3, the first object proposed in the task plan is picked and removed/placed. These stages are repeated until the task is achieved.   Experimental study is provided to compare their method of prompting with prior works, showing an improved performance on the task of Open-Ended Grounding. They then compare their method with VLM-based baseline and an LLM based baseline.   Ablation study is shown to verify the importance of each component in their proposed framework.

**Summary Of Recommendation:**

Overall, this is good work and is among the first set of papers to tightly integrate LVLMs with grasping for zero-shot open-ended tasks. However, the experimental sections need improved presentation, such as fixing missing references to the tables and organizing the content for smoother conceptual flow.

---

### Author Rebuttal · Authors · 2024-08-09

We upload a zip file containing five pdfs with extra material that aims to address reviewer's points. In particular, we included:
1. `reb_SayCanIM.pdf`. We provide detailed analysis and illustrate prompts & in-context examples for our SayCan-IM baseline, incl. discussion about the role of the LLM in this baseline compared to OWG. This is meant to address point by reviewer `Adjp`.
2. `reb_3.3.pdf`. We provide a brief description of our visual prompt design, in order to explicitly mention the components ablated in Sec.4.3. This is meant to address points by reviewers `Adjp`, `hpr3` and `PGH6`.
3. `rebSAM.pdf`. We conducted additional experiments with SAM to demonstrate its representative performance in our OCID-VLG experiments of Sec.4.3. We provide the modified configuration and updated results and visualizations with the updated SAM model. This is meant to address points by reviewers `Adjp, PGH6`.
4. `rebExps.pdf`'. We conducted additional robot experiments trials for the ablation experiments of our Tab.4., as requested by reviewer `PGH6`.
5. `rebRelated.pdf`. We provide additional comparisons of our proposed OWG with other similar semantics-informed grasping works (with a special focus in comparing with GraspGPT), in order to improve our related work section, as requested by reviewer `Adjp`.

---

### Decision · Program_Chairs · 2024-09-04

**Decision:**

Accept

**Comment:**

Update:
The authors responded well to reviewer feedback. While there are still some lingering concerns about the paper novelty, I find the paper's execution and results convincing, and it does show a novel approach that leverages VLMs to synthesize low-level manipulation actions. As such I recommend accepting the paper.

Original:
While reviewers commend the zero-shot capabilities of the proposed method, they are concerned about the thoroughness of the technical explanations and experimental design. I ask the authors to carefully address all reviewer feedback during the rebuttal period.